# LOTUS domain protein MARF1 binds CCR4-NOT deadenylase complex to post-transcriptionally regulate gene expression in oocytes

Li Zhu[1], Suresh K. Kandasamy[1], Susan E. Liao[1] & Ryuya Fukunaga [1]

Post-transcriptional regulation of gene expression plays an essential role during oocyte maturation. Here we report that *Drosophila* MARF1 (Meiosis Regulator And mRNA Stability Factor 1), which consists of one RNA-recognition motif and six tandem LOTUS domains with unknown molecular function, is essential for oocyte maturation. When tethered to a reporter mRNA, MARF1 post-transcriptionally silences reporter expression by shortening reporter mRNA poly-A tail length and thereby reducing reporter protein level. This activity is mediated by the MARF1 LOTUS domain, which binds the CCR4-NOT deadenylase complex. MARF1 binds *cyclin A* mRNA and shortens its poly-A tail to reduce Cyclin A protein level during oocyte maturation. This study identifies MARF1 as a regulator in oocyte maturation and defines the conserved LOTUS domain as a post-transcriptional effector domain that recruits CCR4-NOT deadenylase complex to shorten target mRNA poly-A tails and suppress their translation.

[1] Department of Biological Chemistry, Johns Hopkins University School of Medicine, 725 North Wolfe Street, 521 A Physiology Building, Baltimore, MD 21205, USA. Correspondence and requests for materials should be addressed to R.F. (email: fukunaga@jhmi.edu)

RNA-binding proteins (RBPs) mediate post-transcriptional gene regulation by determining molecular fates of target RNAs[1–4]. In addition to RNA-binding domains, RBPs often have additional auxiliary domains. These auxiliary domains may function as effector domains for post-transcriptional gene regulation directly through enzymatic activity or indirectly by mediating protein–protein interaction. Identifying these effector domains and their molecular functions is critical to understand the roles of RBPs in post-transcriptional gene regulatory mechanism.

MARF1 is an RBP consisting of one RNA-recognition motif (RRM) followed by several tandem LOTUS domains (Limkain, Oskar, and Tudor containing proteins 5 and 7. Also called OST-HTH) (Fig. 1a). Previous studies showed that mouse MARF1 is required for completion of meiosis in oogenesis by reducing protein and mRNA levels of retrotransposons and a few endogenous genes[5–7]. However, the molecular mechanism by which MARF1 regulates gene expression remains unclear.

LOTUS domains are conserved in bacteria, fungi, plants, and animals[8,9]. In animals, LOTUS domain proteins are expressed almost exclusively in the germline and are implicated in RNA regulation. In Drosophila, these LOTUS domain proteins include Oskar, Tejas (human TDRD5), Tapas (human TDRD7), and MARF1 (Meiosis Regulator And mRNA Stability Factor 1 = CG17018. Human MARF1)[5–7,10–13]. However, the molecular function of the conserved LOTUS domain is not fully understood.

Here, we studied the biological and molecular functions of Drosophila MARF1 and its LOTUS domains (Fig. 1a). We found that MARF1 is essential for proper oocyte maturation by regulating cyclin protein levels. When tethered to a reporter mRNA, MARF1 caused shortening of reporter mRNA poly-A tail and reduced reporter protein level. This activity was mediated by MARF1 LOTUS domain. Consistent with this finding, we found that MARF1 binds the CCR4-NOT deadenylase complex via its LOTUS domain. Furthermore, we found that MARF1 binds cyclin A mRNA, shortens its poly-A tail, and reduces Cyclin A protein level during oocyte maturation. Thus, we uncovered the biological and molecular functions of Drosophila MARF1 and defined its conserved LOTUS domains as a post-transcriptional effector domain to recruit the CCR4-NOT deadenylase complex to shorten target mRNA poly-A tails and suppress translation of the mRNAs.

## Results

**MARF1 protein is expressed in late-stage oocytes**. We generated a polyclonal anti-Drosophila MARF1 antibody against a recombinant protein fragment of MARF1 (242–411 aa). We examined the MARF1 expression pattern in Drosophila tissues using hand-dissected wild-type ($w^{1118}$) fly samples. We found that MARF1 protein is expressed in stages 12–13 oocytes, stage 14 oocytes, and whole ovaries (Fig. 1b). MARF1 is only weakly expressed in stages 1–11 oocytes and 0–2 h embryos and is not detected in unfertilized eggs, whole male flies, the carcasses of female flies after ovary dissection, or S2 cells. Thus, MARF1 protein is expressed specifically in ovaries, especially in late-stage (stages 12–14) oocytes, and rapidly disappears during oocyte-to-embryo transition.

We tested if MARF1 protein disappearance during the oocyte-to-embryo transition depends on PNG kinase, a critical regulator of the extensive changes in the translatome during oocyte-to-embryo transition[14,15]. MARF1 protein levels in the 0–2 h embryos of the png missense homozygous mutants ($png^{50/50}$ and $png^{1058/1058}$ (ref. [16])) were increased compared with the heterozygous controls ($png^{50/+}$ and $png^{1058/+}$) while MARF1 protein levels in the stage 14 oocytes were unchanged between the homozygous mutants and controls (Fig. 1c). Therefore, png is required for the rapid disappearance of MARF1 protein during oocyte-to-embryo transition.

**Generation of MARF1 mutant fly strains**. To study the biological and molecular functions of Drosophila MARF1, we created two MARF1 mutant alleles by introducing deletions within the MARF1 coding region using a CRISPR/Cas9 genome-editing system (Fig. 1a)[17]. The MARF1null allele has a 241-nt-long deletion introduced at proximal to the N-terminal end of the protein, which produced a premature stop codon, resulting in production of the N-terminal 103aa fragment of MARF1 (Fig. 1a). We consider this allele as null since this short fragment is unlikely to have any functions. The MARF1RL1 (RL1 = RRM and the first LOTUS domain) allele has a 17-nt-long deletion introduced after the first LOTUS domain, producing a premature stop codon. The MARF1RL1 allele produces the N-terminal 412 aa fragment of MARF1, which includes the RRM and the first LOTUS domain (Fig. 1a). Both MARF1null and MARF1RL1 homozygous mutant flies were viable, showing that MARF1 is dispensable for fly viability.

To validate both the MARF1 mutant fly strains and the antibody, we performed Western blots using the ovary lysates from MARF1 mutant flies and the anti-MARF1 antibody. Full length MARF1 protein was detected in ovary lysates from wild-type (+/+) and heterozygous mutant controls of MARF1null and MARF1RL1 (null/+ and RL1/+) (Fig. 1d). Full-length MARF1 protein was not detected in either homozygous mutants of MARF1null (null/null or −/−) or MARF1RL1 (RL1/RL1). A shorter protein band corresponding to the MARF1 N-terminal 412 aa fragment was detected in heterozygous (RL1/+) and homozygous (RL1/RL1) MARF1RL1 mutants as expected. These results validate our MARF1 mutant fly strains and anti-MARF1 antibody.

**MARF1 is essential for female fertility**. Ovary-specific MARF1 expression (Fig. 1b) suggests that MARF1 plays an important role in oocytes. To test this, we performed fertility assays to determine whether MARF1 is required for fertility (Fig. 2). Male fertility of MARF1null (null/null) was similar to controls (+/+ and null/+) (Fig. 2a), revealing that MARF1 is not crucial for male fertility. This is consistent with the observation that MARF1 protein is not expressed in male flies (Fig. 1b). MARF1null females produced a similar number of eggs to control flies (Fig. 2b). However, none of the eggs laid by MARF1null females hatched while more than 75% of eggs laid by control females hatched (Fig. 2c), revealing that MARF1 is essential for female fertility.

To confirm that female sterility is due to loss of MARF1, we tested if transgenic MARF1 expression could rescue female fertility. Germline-specific expression of transgenic MARF1 (MAT15Tub-Gal4 → UASP-3xHA-MARF1), which enabled expression of transgenic MARF1 protein at a physiological level (Fig. 1d), fully rescued female fertility of MARF1null flies (Fig. 2c). These findings show that expression of MARF1 in germline cells is crucial for female fertility, but not for fecundity.

We also made a transgenic fly expressing EGFP-MARF1 protein to study MARF1 localization. Germline-specific expression of transgenic EGFP-MARF1 (MAT15Tub-Gal4 → UASP-EGFP-MARF1) rescued MARF1null fertility (Supplementary Fig. 1A, B), showing that this transgene is functional. We examined EGFP-MARF1 subcellular localization in oocytes. EGFP-MARF1 was observed in both cytoplasm and nucleus without showing particular subcellular localization (Supplementary Fig. 1D).

By crossing control and MARF1null female flies with male flies expressing a GFP-fused sperm protein, Don Juan (DJ) protein (DJ-GFP), we found that sperms can enter eggs laid by

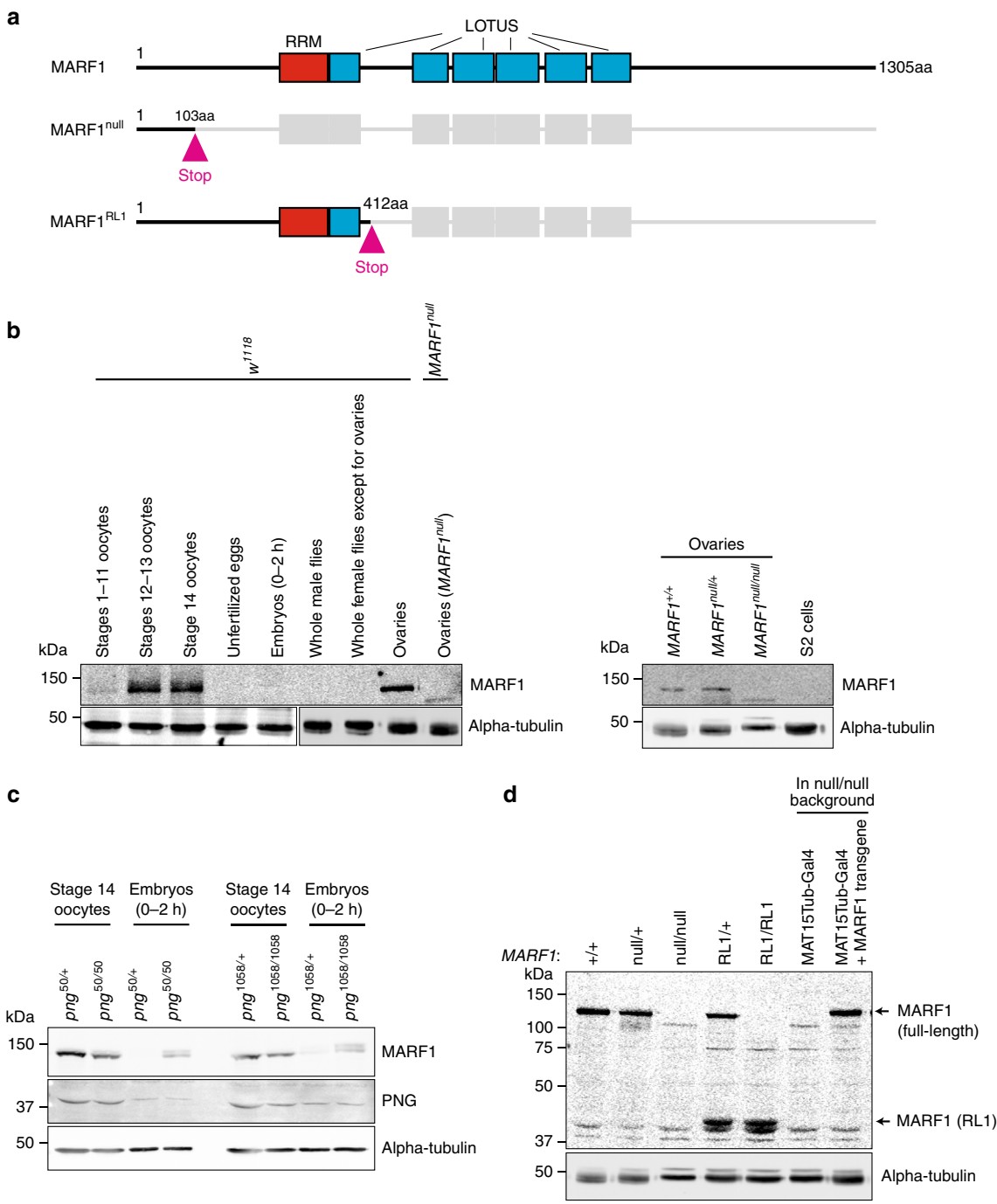

**Fig. 1** MARF1 domain structure, mutant alleles, and protein expression pattern. **a** Domain structures of full-length *Drosophila* MARF1, MARF1[null], and MARF1[RL1]. **b** MARF1 Western blots of dissected oocytes/tissues from control $w^{1118}$ flies, ovaries from *MARF1[null]* (=*MARF1[nlul/null]*) and *MARF1[null/+]*, and S2 cells. **c** Western blots of dissected oocytes and embryos from *png* missense homozygous mutant flies (*png[50/50]* and *png[1058/1058]*) and their heterozygous control flies (*png[50/+]* and *png[1058/+]*). No difference in PNG protein level were observed between homozygous mutant and heterozygous control flies. **d** Western blots of ovary lysates. Transgenic MARF1 protein, which was expressed from the transgene UASP-3xHA-*MARF1* driven by the germline-specific MAT15Tub-Gal4 driver, showed a physiological level of expression

*MARF1[null]* females (we call these eggs as *MARF1[null]* eggs) similar to control eggs (Supplementary Fig. 2A). This result shows that *MARF1[null]* female sterility (Fig. 2) is not due to a loss of sperm entry into *MARF1[null]* eggs.

**MARF1 is required for late-stage oocyte maturation.** In *Drosophila*, meiosis prophase I arrest occurs in stages 6–11 oocytes,

oocyte maturation occurs in stages 12–13 oocytes, and a stable meiotic metaphase I arrest occurs in stage 14 oocytes ( = mature oocytes)[18]. The meiotic metaphase I arrest is released as stage 14 oocytes traverse through oviduct, regardless of whether they have been fertilized. Thus, meiosis is completed in both unfertilized eggs and fertilized embryos. Following primary meiotic arrest in stages 6–11, oocytes are transcriptionally quiescent until zygotic

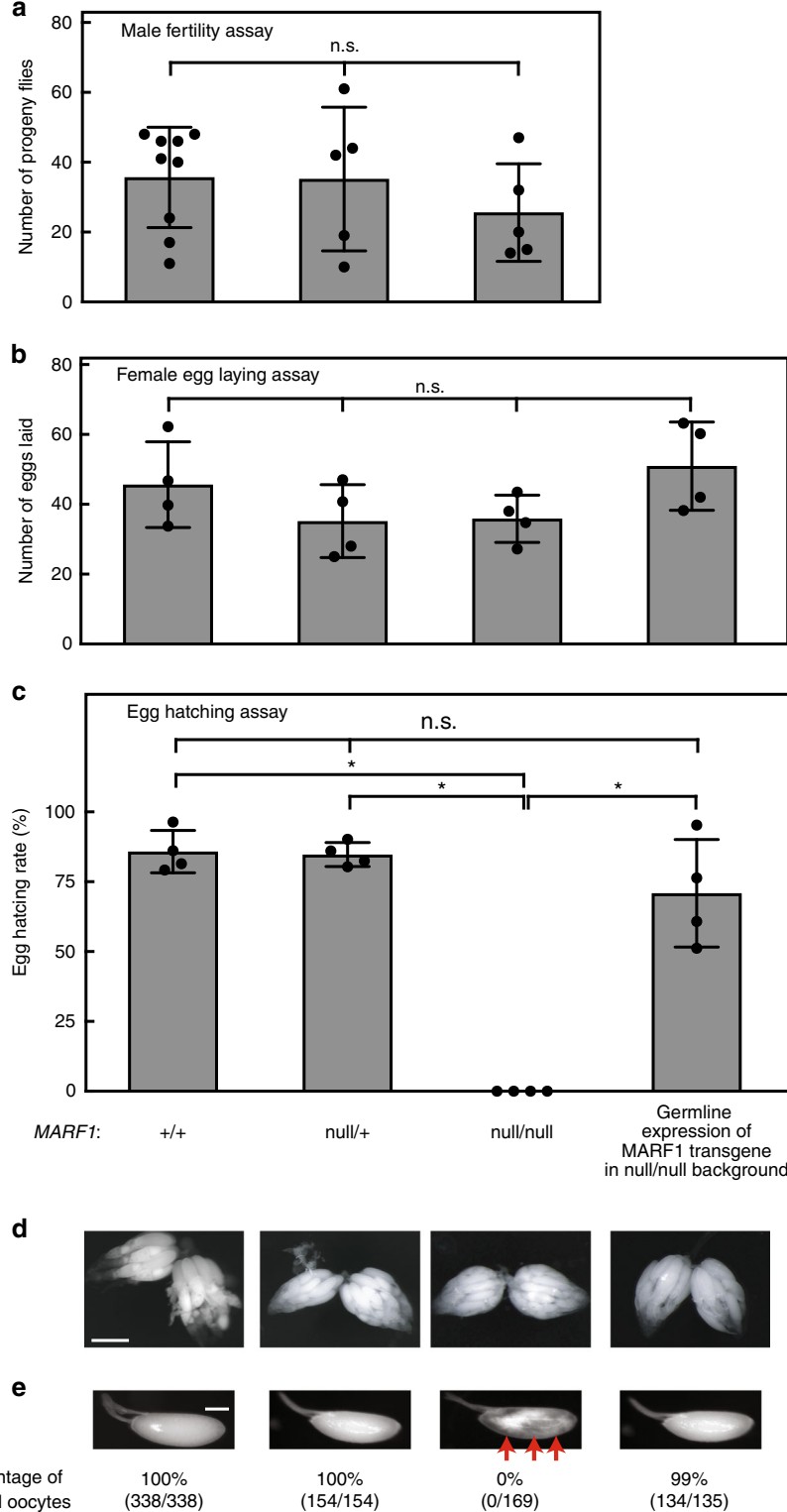

**Fig. 2** *MARF1null* female flies are sterile and have abnormal stage 14 oocytes. **a** Male fertility assay. Numbers of progeny flies from crosses between test males and OregonR wild-type females are shown. Mean ± SD ($n = 10$). **b**, **c** Female fertility assays. **b** The numbers of eggs laid by test females crossed with OregonR wild-type males and **c** the hatching rates of the eggs. Mean ± SD ($n = 3$). *P*-value < 0.05 (Student's *t*-test) are indicated by *. **d** Stereomicroscope images of dissected whole ovaries. Scale bar is 500 μm. **e** Stereomicroscope images of dissected stage 14 oocytes. Abnormal yolk distribution in *MARF1null* stage 14 oocyte is indicated by red arrows. The percentages of normal oocytes are shown. The numbers of normal oocytes/the numbers of observed oocytes are shown in parenthesis. Germline expression of the MARF1 transgene (MAT15Tub-Gal4 → UASP-3xHA-MARF1) in the *MARF1null* background rescued hatching rate and yolk distribution in stage 14 oocytes. Scale bar is 100 μm

transcription starts in embryos several hours after fertilization. During this period of time, gene expression is regulated exclusively at the post-transcriptional level.

Despite *MARF1^null^* females laying a comparable number of eggs compared to controls, eggs laid by *MARF1^null^* females fail to develop into viable progeny (Fig. 2b, c). This may be due to defective oocyte maturation or defects during egg activation. While gross ovary structure and morphologies of earlier stage oocytes (stages 1–11) of *MARF1^null^* flies appear normal (Fig. 2d), the yolk distribution within *MARF1^null^* stage 14 oocytes is abnormal with aggregation (Fig. 2e, indicated by red arrows). Yolk distribution in *MARF1^null^* stage 14 oocytes was rescued by germline expression of transgenic MARF1 (Fig. 2e and Supplementary Fig. 1C). This shows that MARF1 is required for proper oocyte maturation.

*MARF1^null^* stage 14 oocytes exhibited proper metaphase I arrest as in control stage 14 oocytes as observed by proper formation of a meiotic spindle (Fig. 3a). In control *w^1118^* unfertilized eggs, a polar body was observed, indicating completion of Meiosis I and II (Fig. 3b). However, no meiotic spindles or polar bodies were observed in *MARF1^null^* unfertilized eggs. Thus *MARF1^null^* stage 14 oocytes are abnormal, meiosis does not complete properly, and meiotic spindles are disintegrate after ovulation, resulting in complete sterility. We conclude that MARF1 plays a crucial role in late-stage oocyte maturation and meiotic progression.

*MARF1^RL1^* flies exhibited similar fertility and oocyte phenotypes as *MARF1^null^* flies (Supplementary Fig. 2B–E), suggesting that MARF1 1–412 aa fragment cannot substitute physiological functions of full-length MARF1. We confirmed this by testing whether germline expression of transgenic MARF1 1–411 aa fragment or 1–340 aa fragment can rescue female fertility and stage 14 oocyte morphology of *MARF1^null^*. Neither partial fragment rescued female fertility or stage 14 oocyte morphology as full-length transgenic MARF1 did (Supplementary Fig. 2F–H). Thus, the multiple tandem LOTUS domains and/or the C-terminal region, not just the RRM and the first LOTUS domain, are required for MARF1 function.

## MARF1 is required for germplasm protein localization.

A key feature of *Drosophila* oocyte maturation is establishment of the posterior germplasm which gives rise to primordial germ cells after fertilization. The germplasm includes specific RNAs and proteins, including DEAD-box RNA helicase Vasa and Piwi protein Aubergine (Aub)[19]. In control oocytes, both Vasa and Aub localize to the posterior of stage 10 and stage 14 oocytes (Fig. 4a, c, indicated by red arrows). While Vasa and Aub showed proper posterior localization in *MARF1^null^* stage 10 oocytes, Vasa and Aub did not exhibit posterior (germplasm) localization in *MARF1^null^* stage 14 oocytes (Fig. 4b, d, indicated by a red arrow). Thus, posterior localizations of germplasm proteins are lost in *MARF1^null^* stage 14 oocytes.

## MARF1 is required for regulation of meiotic cyclin levels.

Cyclin protein levels are dynamically and finely regulated during oocyte maturation. Of particular importance are Cyclins A, B, and B3, which bind cyclin-dependent kinase 1 (CDK1) to regulate its kinase activity and substrate specificity, which controls meiotic cell cycle progression.

In *MARF1^null^* late-stage oocytes, we found that Cyclin A protein level was increased whereas Cyclin B and Cyclin B3 protein levels were decreased compared to control oocytes (Fig. 5 and Supplementary Fig. 3A). In controls, Cyclin A protein level was moderate in stages 1–11 oocytes, highest in stages 12–13 oocytes, and very low in stage 14 oocytes (Fig. 5). Cyclin A

protein level was significantly increased in *MARF1^null^* stages 12–13 oocytes and 14 oocytes than in controls. In controls, Cyclin B and Cyclin B3 protein levels became detectable in stages 12–13 oocytes and highest in stage 14 oocytes. However, they remained undetectable in *MARF1^null^* stages 12–13 and stage 14 oocytes. *MARF1^RL1^* stage 14 oocytes exhibited similar dysregulation of cyclin protein levels (Supplementary Fig. 3B).

CDK1 protein level was lower in *MARF1^null^* and *MARF1^RL1^* stage 14 oocytes than in control stage 14 oocytes (Fig. 5 and Supplementary Fig. 3). During mitosis, CDK1 Thr14, Tyr15, and Thr161 residues are post-translationally phosphorylated. Phosphorylation of Thr14 and Tyr15 inhibits CDK1 kinase activity while Thr161 phosphorylation promotes CDK1 kinase activity during mitosis cell cycle[20]. Whether these same residues are phosphorylated during meiosis are unknown. We did not detect phosphorylation at CDK1 Thr14 or Thr161 in control or *MARF1* mutant stage 14 oocytes (Supplementary Fig. 3B). We found that CDK1 Tyr15 was phosphorylated in control stage 14 oocytes and unfertilized eggs, but not in *MARF1^null^* stage 14 oocytes and unfertilized eggs (Fig. 5 and Supplementary Fig. 3A) or in *MARF1^RL1^* stage 14 oocytes (Supplementary Fig. 3B).

The anti-phospho-Ser/Thr-Pro MPM-2 antibody, which detects cell cycle-regulated phosphoproteins, many of which are phosphorylated by CDK1[21], showed stage-specific phosphoprotein expression patterns in control oocytes (Fig. 5a). Phosphoprotein profiles in *MARF1^null^* stages 1–11 and stages 12–13 oocytes were similar to controls. In contrast, phosphoprotein profiles were altered in *MARF1^null^* stage 14 oocytes and more dramatically in *MARF1^null^* unfertilized eggs, compared with controls (Fig. 5a and Supplementary Fig. 3A).

Germline-specific expression of transgenic MARF1 protein rescued protein levels of cyclins and CDK1 and phosphoprotein profile in *MARF1^null^* stage 14 oocytes (Supplementary Fig. 3A).

No major changes in either Vasa or Aub protein levels were observed in any stages between control and *MARF1* mutant oocytes (Fig. 5 and Supplementary Fig. 3B). These results reveal that observed loss of Vasa and Aub germplasm localization in *MARF1^null^* stage 14 oocytes (Fig. 4) is not due to a change in amount of Vasa and Aub proteins. Cyclin D, Twine (a phosphatase that dephosphorylates phosphorylated Tyr15 of CDK1. CDC25), Cort (a member of the Cdc20 family of activators of the Anaphase Promoting Complex), Belle, Nanos, and Oskar protein levels were not changed between control and *MARF* mutant stage 14 oocytes (Supplementary Fig. 3A, B). These results show that dysregulation of protein levels of cyclins (Cyclin A, B and B3) and CDK1 in *MARF1* mutant oocytes is specific to these proteins.

Furthermore, *cyclin A*, *cyclin B*, *cyclin B3* and *cdk1* mRNA levels were not changed in *MARF1^null^* oocytes compared with control oocytes (Fig. 6), indicating that post-transcriptional dysregulation underlies their altered protein levels in *MARF1^null^* oocytes. These results show that *MARF1* is required for post-transcriptional regulation of cyclin protein levels and for protein level, Tyr15 phosphorylation status, and phosphorylation activities, of CDK1.

## Cyclin A degradation is unchanged in *MARF1^null^* oocytes.

In controls, Cyclin A protein is most highly expressed in stages 12–13 oocytes, but it is undetectable in stage 14 oocytes (Fig. 5), indicating that it is actively degraded in stage 14 oocytes. Increased Cyclin A protein levels observed in MARF1 mutant oocytes compared with control oocytes (Fig. 5) can be caused by either increased Cyclin A protein production or decreased Cyclin A degradation. To test whether the Cyclin A protein degradation

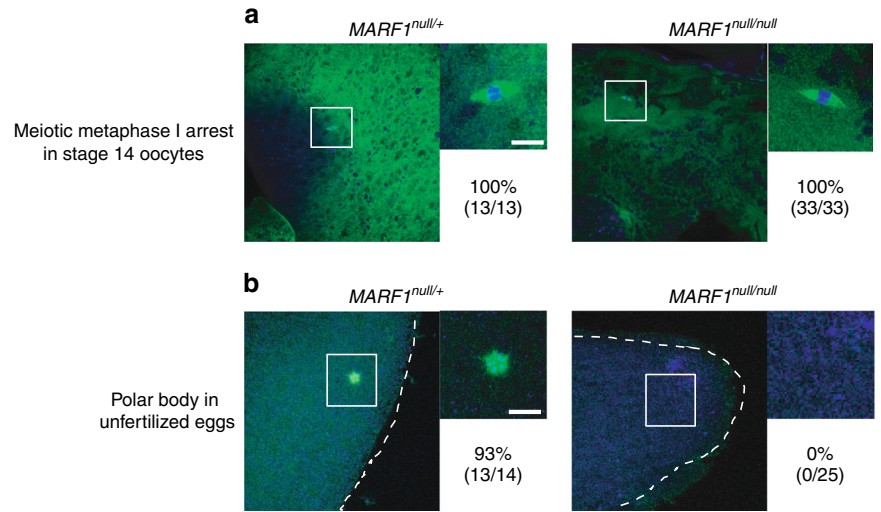

**Fig. 3** $MARF1^{null}$ eggs fail to complete meiosis. **a** Meiotic spindles show proper meiotic metaphase I arrest in control and $MARF1^{null}$ stage 14 oocytes. The percentages of normal oocytes with proper meiotic spindle are shown. The numbers of normal oocytes with proper meiotic spindle/the numbers of observed oocytes are shown in parenthesis. Scale bar is 5 μm. **b** Polar body shows completion of meiosis in control unfertilized eggs, but not in $MARF1^{null}$ unfertilized eggs. Neither meiotic spindle nor polar body were observed in $MARF1^{null}$ unfertilized eggs. The percentages of normal unfertilized eggs with a proper polar body are shown. The numbers of normal unfertilized eggs with a proper polar body/the numbers of observed unfertilized eggs are shown in parenthesis. Scale bar is 5 μm. DNA was stained with DAPI (blue) and alpha-Tubulin was visualized with FITC-conjugated anti-alpha-Tubulin antibody (green)

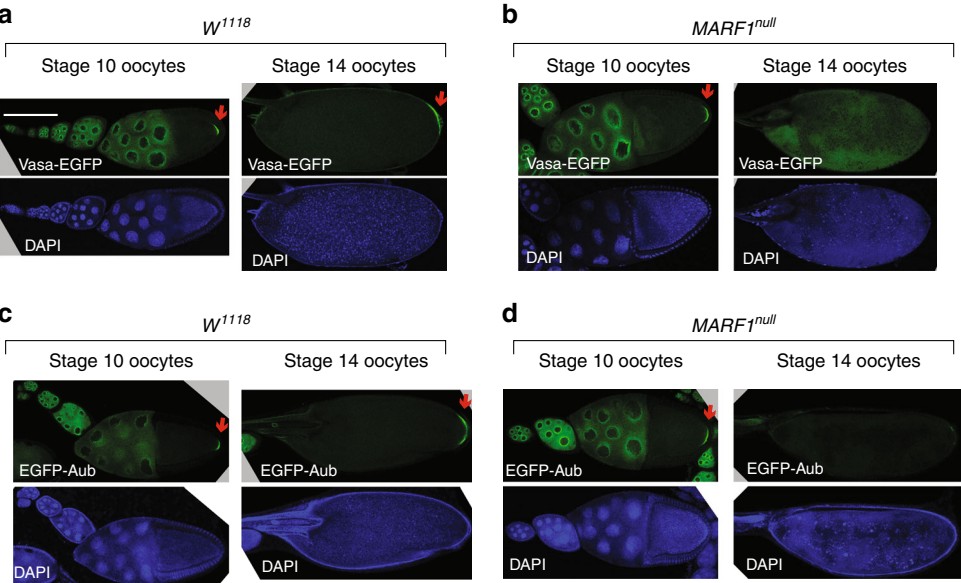

**Fig. 4** Loss of germplasm localization of Vasa and Aub in $MARF1^{null}$ stage 14 oocytes. **a** Vasa-EGFP localization in control $w^{1118}$ stage 10 and stage 14 oocytes. **b** Vasa-EGFP localization in $MARF1^{null}$ stage 10 and stage 14 oocytes. **c** EGFP-Aubergine localization in control $w^{1118}$ stage 10 and stage 14 oocytes. **d** EGFP-Aubergine localization in $MARF1^{null}$ stage 10 and stage 14 oocytes. Scale bar is 100 μm

rate is changed in $MARF1^{null}$ oocytes compared with control oocytes, we measured degradation rates of recombinant HA-Cyclin A protein added to $MARF1^{null}$ and control ($MARF1^{null/+}$) stage 14 oocyte lysates in vitro (Supplementary Fig. 4). Consistent with active degradation of Cyclin A in control stage 14 oocytes in vivo, recombinant HA-Cyclin A protein was degraded over time while endogenous alpha-Tubulin remained stable in our in vitro assay. Recombinant HA-Cyclin A was degraded at similar rates in $MARF1^{null}$ and control lysates with half-lives ($t^{1/2}$) of $2.5 \pm 0.4$ h and $2.5 \pm 0.6$ h (mean ± SD, $n = 3$), respectively (Supplementary Fig. 4C). These results indicate that the increased Cyclin A protein level in $MARF1^{null}$ oocytes (Fig. 5) is caused by

an increased production of Cyclin A protein, rather than a decreased degradation rate.

**MARF1 tethering to a reporter mRNA causes poly-A shortening.** To examine effects of MARF1 binding to mRNA, we tethered transgenic MARF1 protein to a reporter GFP mRNA in oocytes using a lambdaN peptide-BoxB hairpin RNA system (Fig. 7a)[22]. LambdaN-HA-MARF1 is tethered to the reporter GFP mRNA by the binding of lambdaN peptide to the BoxB hairpins in the 3′ UTR of the reporter GFP mRNA. We also tested tethering of lambdaN-HA peptide only and lambdaN-HA-Piwi as negative controls and lambdaN-HA-GW182 as a positive

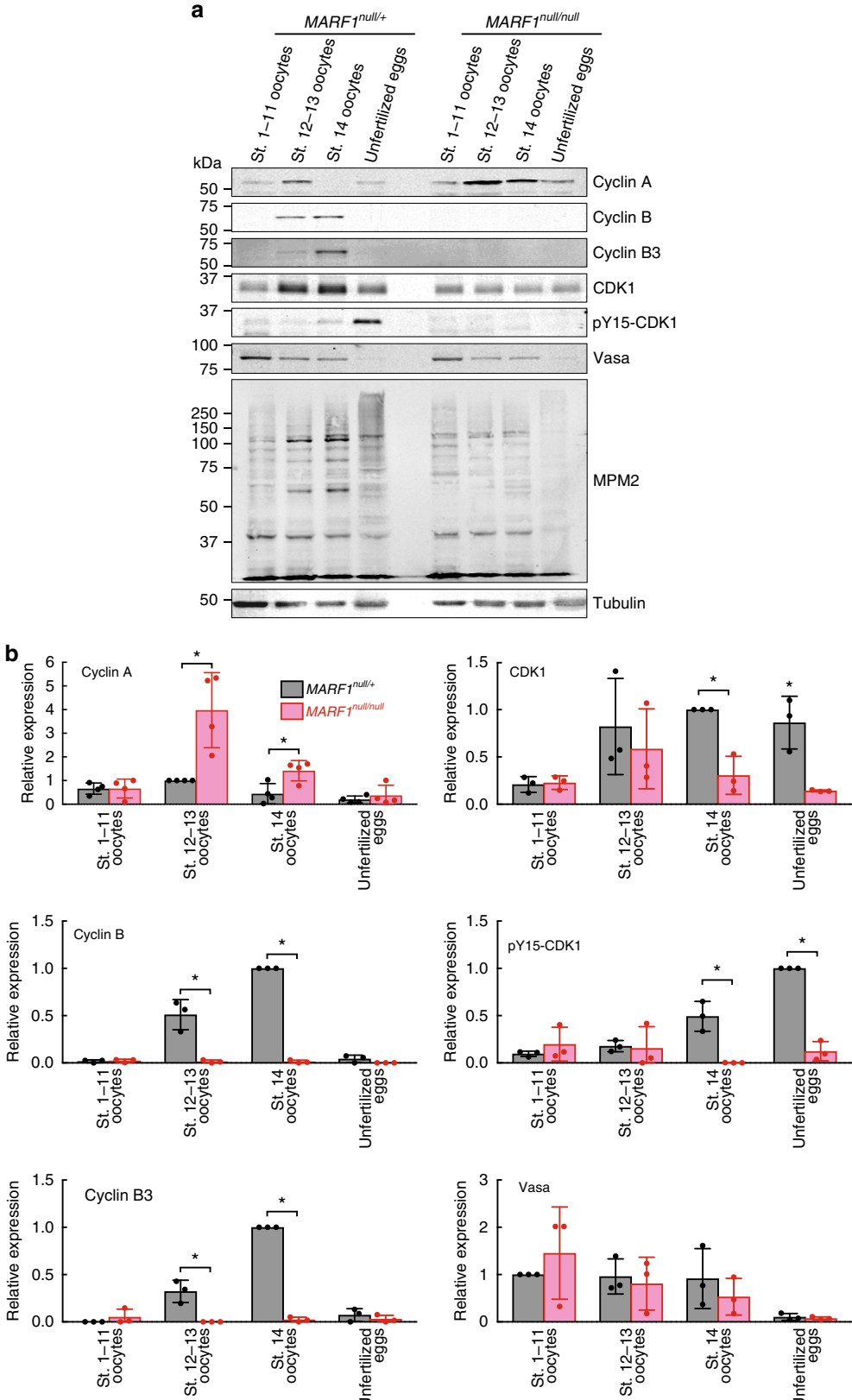

**Fig. 5** Dysregulation of meiotic cyclin protein levels and phosphoprotein profile in *MARF1null* stage 14 oocytes. **a** Western blots using dissected oocytes and unfertilized eggs. **b** Western blot quantification, whose representative images are shown in **a**. Protein levels were normalized to that in control (*MARF1null/+*) oocyte stage showing the highest expression (Stages 1–11 for Vasa. Stages 12–13 for Cyclin A. Stage 14 for Cyclin B, Cyclin B3, and CDK1. Unfertilized eggs for pY15-CDK1). Mean ± SD ($n = 4$ for CycA and $n = 3$ for all the others). *P*-value < 0.05 (Student's *t*-test) are indicated by *

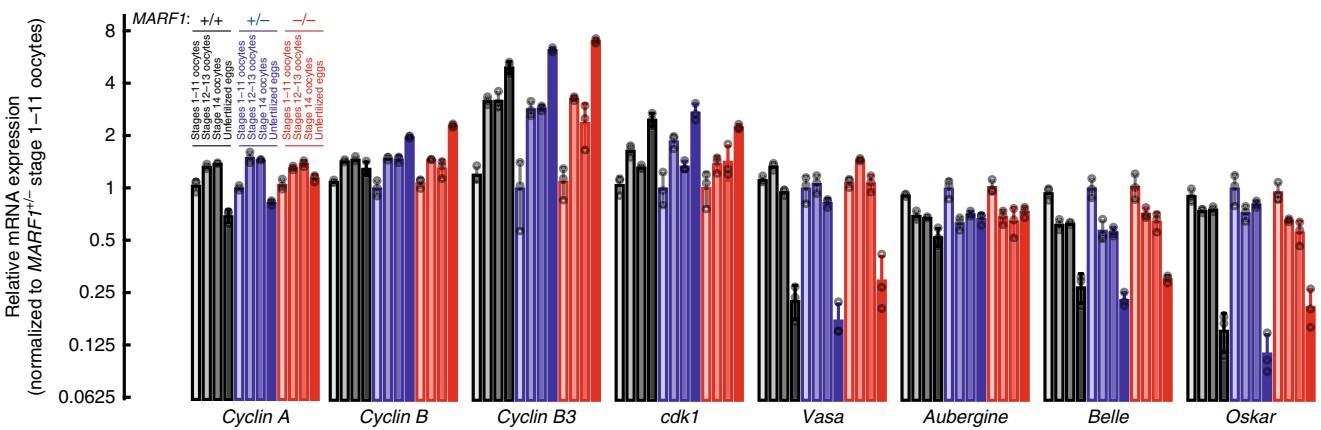

**Fig. 6** *Cyclin* mRNA levels are unchanged *in MARF1^null^ stage 14 oocytes*. mRNA levels relative to *MARF1^null/+^* stages 1–11 oocytes determined by mRNA-seq. Mean ± SD ($n = 3$)

control for post-transcriptional suppression[22]. Tethered Piwi does not affect protein levels produced from a reporter mRNAs containing BoxB hairpins[22]. GW182 is a known translational suppressor[23–25]. Tethered GW182 represses translation and destabilizes mRNAs in S2 cells while it represses translation without changing mRNA levels in oocytes[22]. We expressed these lambdaN-HA-fused transgenic proteins in oocytes using maternal-alpha-tubulin-Gal4 driver in the GFP reporter background. GFP protein was well expressed in lambdaN-HA-control and lambdaN-HA-Piwi stage 14 oocytes as observed by confocal microscopy imaging and Western blotting (Fig. 7b–d). In contrast, GFP protein level was strongly decreased in lambdaN-HA-GW182 stage 14 oocytes as expected. Interestingly, GFP protein level was also strongly decreased in lambdaN-HA-MARF1 stage 14 oocytes (Fig. 7b–d). Thus, MARF1, when tethered to the 3′ UTR of a reporter mRNA, decreases the protein level produced from the mRNA.

To understand the molecular mechanism by which tethered MARF1 reduces reporter protein level, we examined the poly-A tail length of GFP reporter mRNA using an extension Poly-A Test (ePAT) assay[26]. We found that MARF1 tethering caused shortening of GFP reporter mRNA poly-A tail length similar to GW182 tethering (Fig. 7e, f). In contrast, the poly-A tail length of a control mRNA (*cdk1* mRNA), which was not tethered by lambdaN-HA-proteins, was not changed (Fig. 7f), confirming that the poly-A shortening of the GFP mRNA is due to the lambdaN-HA-MARF1 binding. Next, we measured the abundance of the GFP reporter mRNA by qRT-PCR. Neither tethered MARF1 nor tethered GW182 changed GFP reporter mRNA level compared to control (Fig. 7g). These results indicate that MARF1 post-transcriptionally decreases reporter protein level by shortening reporter mRNA poly-A tails.

Tethering of the MARF1 1–411 aa fragment, which includes the RRM and the first LOTUS domain, reduced GFP protein level as the full-length MARF1 did (Fig. 8). In contrast, tethering of the MARF1 1–340 aa fragment, which contains the RRM but not LOTUS domain, did not reduce GFP protein level. These results reveal that the LOTUS domain is required to suppress the reporter protein level.

**MARF1 LOTUS domain binds the CCR4-NOT deadenylase complex**. Considering that the MARF1 amino acid sequence does not have any homology to nucleases, we hypothesized that MARF1 binds and recruits a deadenylase complex to target mRNAs. In *Drosophila*, there are two distinct deadenylase complexes: the CCR4-NOT complex and the Pan2-Pan3 complex[27]. To determine whether MARF1 binds either of these

deadenylase complexes, we performed co-immunoprecipitation using fly ovary lysates with or without germline expression of 3xHA-MARF1 (MAT67Tub-Gal4 → UASP-3xHA-*MARF1*) followed by Western blot. We found that 3xHA-MARF1 co-immunoprecipitated with CCR4-NOT deadenylase complex components (Not1, Not2, Not3, and Caf1) in an RNA-independent manner (Fig. 9a). 3xHA-MARF1 did not co-immunoprecipitate with endogenous MARF1, DEAD-box helicase Vasa, DEAD-box helicase Belle, DEAD-box helicase Me31B, Oskar, GW182, Ge-1, Cyclin A, Cyclin B, Cyclin B3, CDK1, or alpha-Tubulin, showing that the interaction between MARF1 and the CCR4-NOT deadenylase complex is specific (Supplementary Fig. 5A).

As no interaction was observed between transgenic 3xHA-MARF1 and endogenous MARF1, MARF1 protein does not form a homodimer or oligomer. Furthermore, as no interaction was observed between 3xHA-MARF1 and Cyclin A, Cyclin B, Cyclin B3, or CDK1, dysregulation of these proteins levels in *MARF1* mutant oocytes (Fig. 5 and Supplementary Fig. 3) is not due to disruption of a protein–protein interaction with MARF1.

We tested whether endogenous MARF1 binds the endogenous CCR4-NOT deadenylase complex. We performed anti-Not1 co-immunoprecipitation using wild-type (*w^1118^*) fly ovary lysates. Endogenous MARF1 was co-immunoprecipitated with endogenous Not1 while they were not immunoprecipitated when IgG was used as a negative control in immunoprecipitation (Fig. 9b). Endogenous Not1 did not co-immunoprecipitate with Vasa and alpha-Tubulin, showing the specific interaction between endogenous MARF1 and the endogenous CCR4-NOT deadenylase complex. Moreover, we found that transgenic CCR4-HA co-immunoprecipitated with endogenous MARF1 (Supplementary Fig. 5B).

Next, we tested whether MARF1 binds Pan2 and Pan3, components of the Pan2–Pan3 deadenylase complex, and found that they did not co-immunoprecipitate with MARF1 (Supplementary Fig. 5C, D). Together, these results reveal that MARF1 specifically binds the CCR4-NOT deadenylase complex.

We tested the MARF1 1–340 aa fragment and the MARF1 1–411 aa fragment in the co-immunoprecipitation experiments (Fig. 9c). Full-length MARF1 and the MARF1 1–411 aa fragment co-immunoprecipitated with Not1, but not with Vasa or alpha-Tubulin (Fig. 9c). In contrast, the MARF1 1–340 aa fragment did not co-immunoprecipitate with Not1. Since the difference between these two fragments is the presence of the first LOTUS domain in the MARF1 1–411 aa fragment, these results indicate that the LOTUS domain binds the CCR4-NOT deadenylase complex. This is consistent with the finding that the LOTUS

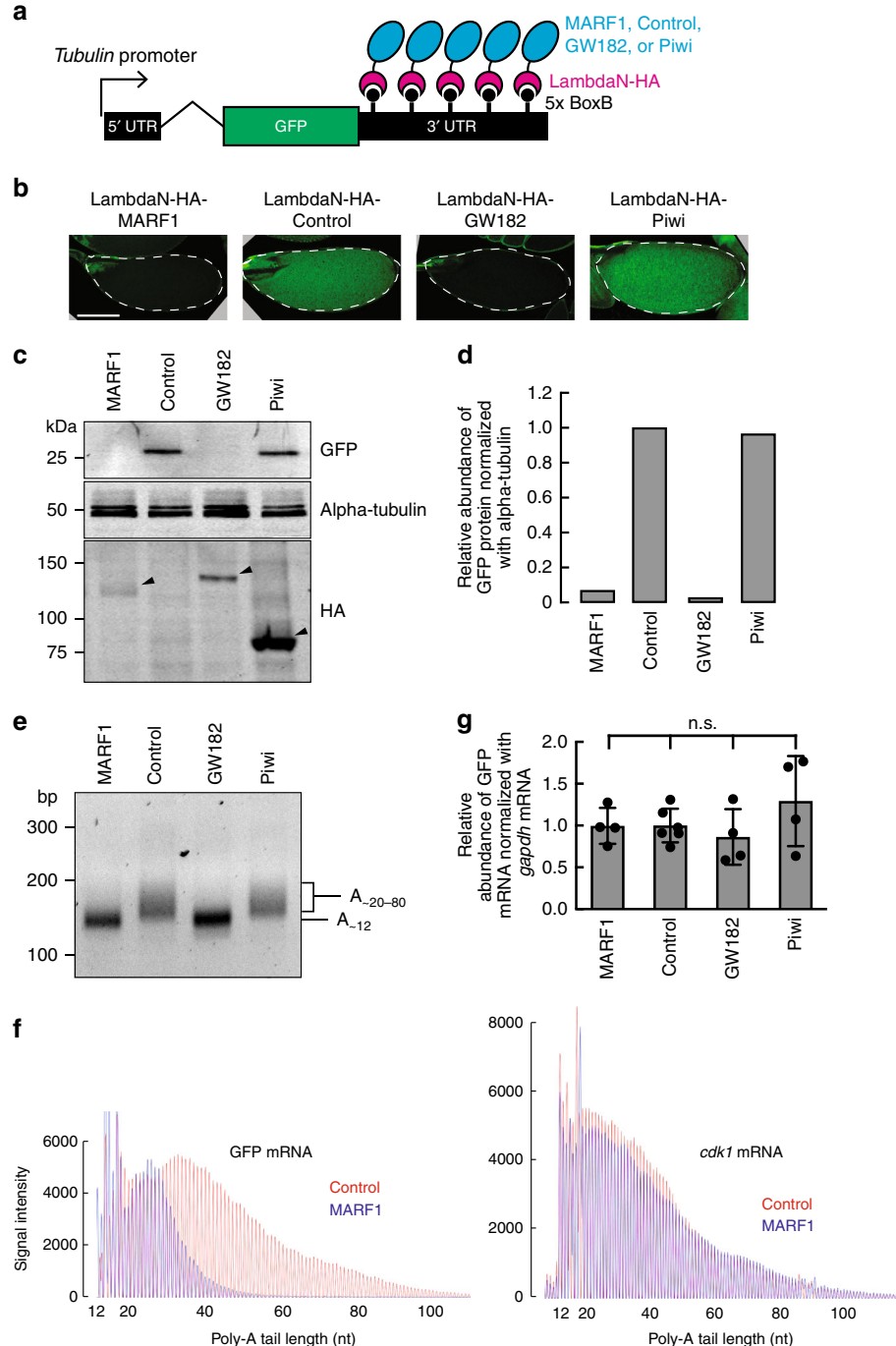

**Fig. 7** Tethered MARF1 shortens reporter mRNA poly-A tail and reduces reporter protein level. **a** GFP-5x BoxB reporter structure, harboring a ubiquitous tubulin promoter, GFP-coding sequence, and a 3' UTR containing five BoxB hairpins. LambdaN-HA-fused control peptide, MARF1, GW182, and Piwi under a UASP promoter were expressed in germline cells using maternal-alpha-tubulin-Gal4 driver. **b** Confocal images of GFP signal in stage 14 oocytes. Scale bar is 100 μm. **c** Western blots using stage 14 oocyte lysates. Black triangles indicate the transgenic lambda-HA-fused proteins. **d** Quantification of band intensities in **c**. **e** ePAT assay measuring GFP reporter mRNA poly-A tail length in stage 14 oocytes. The amplified DNA sizes were analyzed on an agarose gel. **f** ePAT assay measuring poly-A tail lengths of GFP reporter mRNA and a negative control *cdk1* mRNA in stage 14 oocytes. Amplified DNA sizes were analyzed by capillary fragment analysis. **g** Relative abundance of GFP-5xBoxB mRNA normalized by *gapdh* mRNA determined by qRT-PCR. Mean ± SD (*n* = 6 for the control and *n* = 4 for all the others)

domain is responsible for the reporter GFP protein suppression in the tethering assay (Fig. 8).

**MARF1 binds *cyclin A* mRNA.** To identify endogenous MARF1 target mRNAs, we performed RNA-co-immunoprecipitation (RIP) followed by qRT-PCR. Transgenic HA-tagged

MARF1 specifically bound *cyclin A* mRNA, but did not bind *cyclin B*, *cyclin B3*, *cdk1*, and *rp49* mRNAs (Fig. 9d).

Cyclin A was the only protein whose level was increased in *MARF1^null* stage 14 oocytes compared with control oocytes (Fig. 5 and Supplementary Fig. 3). MARF1 tethering causes shortening of GFP reporter mRNA poly-A tail (Fig. 7). MARF1 specifically

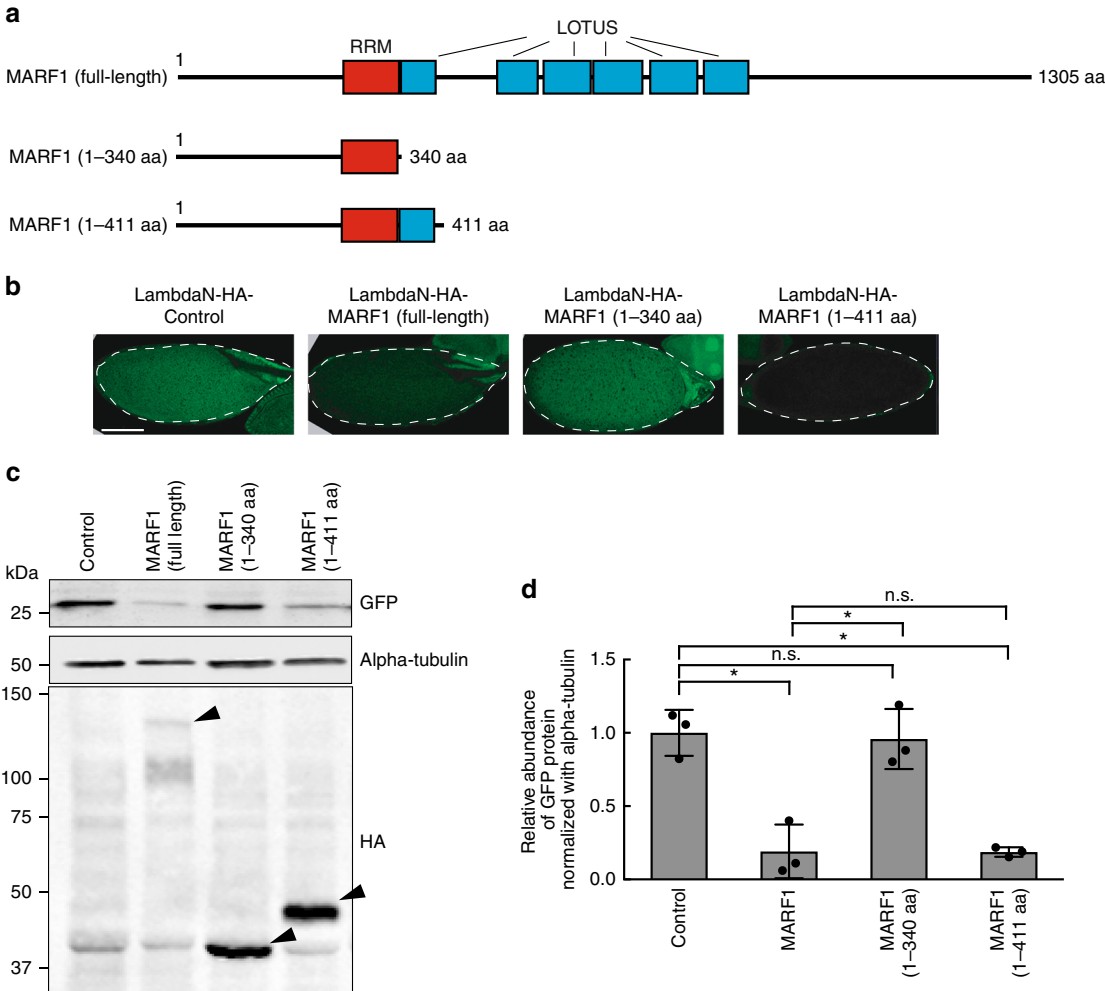

**Fig. 8** MARF1 LOTUS domain mediates target silencing. **a** Full-length MARF1 and MARF1 fragments tested in the GFP-5x BoxB reporter tethering assay. **b** Confocal images of GFP signal in stage 14 oocytes. Scale bar is 100 μm. **c** Western blots using stage 14 oocytes. Black triangles indicate transgenic lambda-HA-fused proteins. **d** Quantification of band intensities in **d**. Mean ± SD ($n = 3$). P-value < 0.05 (Student's t-test) is indicated by *

binds *cyclin A* mRNA (Fig. 9d). Based on these findings altogether, we hypothesized that MARF1 binds *cyclin A* mRNA in stage 14 oocytes, shortens its poly-A tail, and suppresses Cyclin A protein expression. To test this hypothesis, we examined *cyclin A* mRNA poly-A tail length in control and *MARF1^null* stage 14 oocytes using ePAT assay. We found that *cyclin A* mRNA poly-A tail length is longer in *MARF1^null* stage 14 oocytes than in control stage 14 oocytes (Fig. 9e), supporting our hypothesis. In contrast, the poly-A tail length of *cdk1* mRNA, which was not bound by MARF1 (Fig. 9d), was not altered between control and *MARF1^null* stage 14 oocytes (Fig. 9e).

## Discussion

Here, we uncovered the biological and molecular functions of *Drosophila* MARF1, an RBP consisting of one RRM and six LOTUS domains (Fig. 1). We showed that MARF1 is expressed in late-stage oocytes (Fig. 1) and is required for proper oocyte maturation by regulating cyclin protein levels (Figs. 2–5, Supplementary Figs. 1–3). MARF1 binds the CCR4-NOT deadenylase complex via its LOTUS domain (Fig. 9a–c and Supplementary Fig. 5) to shorten target mRNA poly-A tails (Fig. 7e, f) and thus reducing cyclin protein levels (Fig. 7b–d) without changing cyclin mRNA levels (Fig. 7g). Thus, we

define MARF1 LOTUS domain as a post-transcriptional effector domain that binds the CCR4-NOT deadenylase complex.

Recent studies by others showed that the LOTUS domains of *Drosophila* Oskar, Tejas, and Tapas bind germline DEAD-box RNA helicase Vasa to stimulate Vasa ATPase and helicase activities[31–33]. Crystallographic studies showed that the LOTUS domain of Oskar forms a homodimer and that each of the monomer subunits binds the C-terminal domain of the Vasa DEAD-box helicase core on the side opposite to the dimerization interface[33]. In contrast, we showed that the MARF1 LOTUS domain binds the CCR4-NOT deadenylase complex, but does not bind Vasa, Oskar, or another molecule of MARF1 (Fig. 9a, c and Supplementary Fig. 5)[33].

The LOTUS domains found in Oskar, Tejas, and Tapas, but not MARF1, have a C-terminal extension, which is required for interaction with Vasa[33]. Hence the LOTUS domains are divided into two subclasses: (1) extended LOTUS (eLOTUS) domain that is present in Oskar, Tejas, and Tapas, has a C-terminal extension, and binds Vasa, and (2) minimal LOTUS (mLOTUS) domain that is present in MARF1, lacks a C-terminal extension, does not bind Vasa[33], and instead binds the CCR4-NOT deadenylase complex (Fig. 9 and Supplementary Fig. 5). Thus, although eLOTUS and mLOTUS domains share core sequence homology

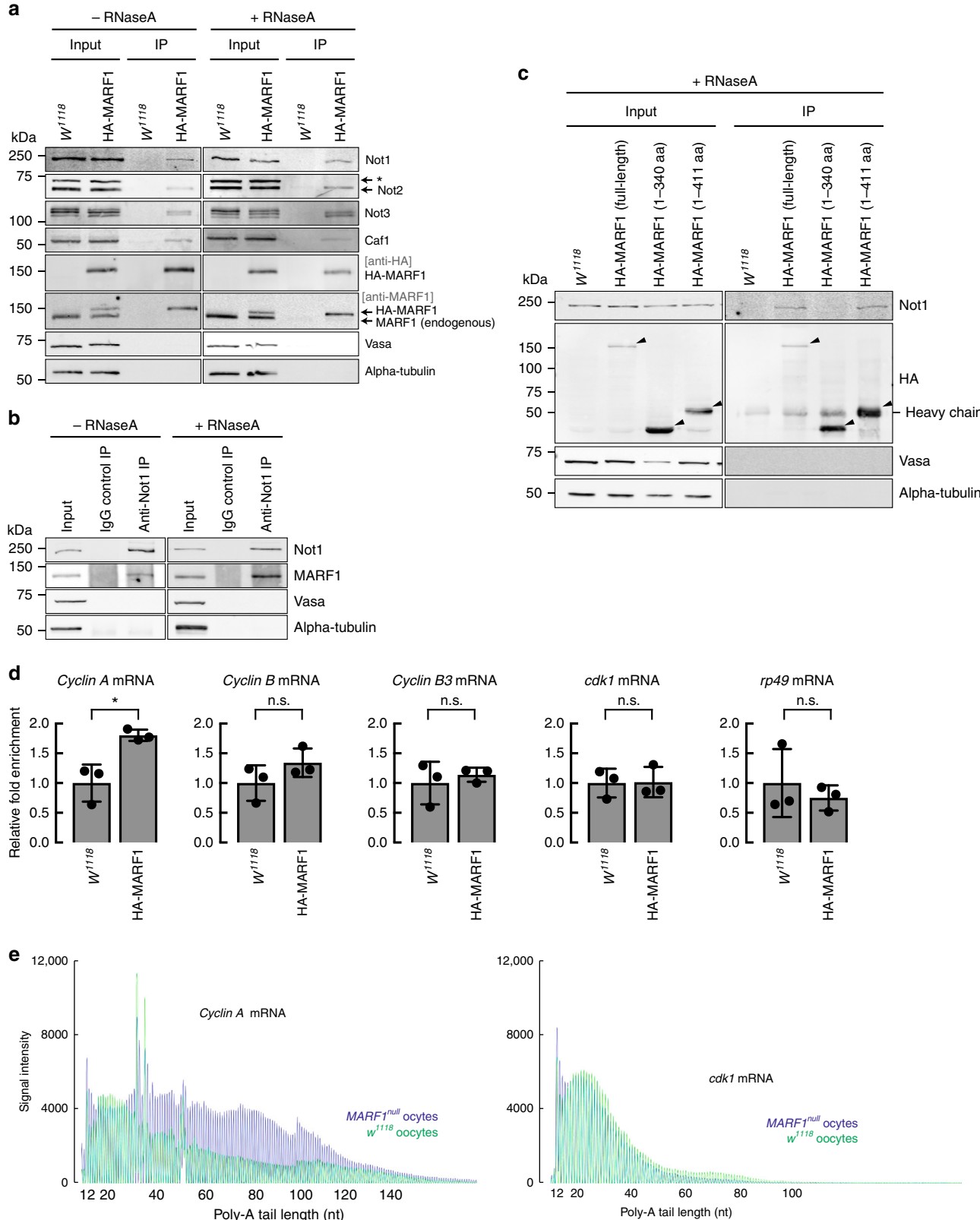

except for the C-terminal extension, they mediate distinct protein–protein interactions. Interestingly, eLOTUS proteins (Oskar, Tejas/TDRD5, Tapas/TDRD7) contain a single eLOTUS domain while mLOTUS proteins (MARF1) contain multiple tandem mLOTUS domains.

We also showed that MARF1 binds *cyclin A* mRNA (Fig. 9d). In *MARF1null* mutant late-stage oocytes, *cyclin A* mRNA poly-A tail is longer (Fig. 9e) and Cyclin A protein level is increased (Fig. 5 and Supplementary Fig. 3), without change in the *cyclin A* mRNA level (Fig. 6). The degradation rate of Cyclin A protein was not

**Fig. 9** MARF1 binds the CCR4-NOT deadenylase complex and *cyclin A* mRNA. **a** Co-immunoprecipitation using anti-HA antibody-conjugated beads followed by Western blotting. Ovary lysates expressing 3xHA-MARF1 (MAT67Tub-Gal4 → UASP-3xHA-MARF1) and those from *w*[1118] negative control were tested. **b** Co-immunoprecipitation using anti-HA antibody-conjugated beads followed by Western blotting. Ovary lysates expressing 3xHA-tagged MARF1 (MAT67Tub-Gal4 → UASP-3xHA-MARF1 full-length or fragments) and those from *w*[1118] negative control were tested. Black triangles indicate the detected 3xHA-tagged MARF1 proteins. **c** Co-immunoprecipitation using ovary lysates from *w*[1118] and anti-Not1 antibody or negative control IgG-bound protein G beads followed by Western blotting. **d** Fold enrichment of mRNAs relative to a control *gapdh* mRNA that were co-immunoprecipitated with MARF1 by anti-HA antibody-conjugated beads and were eluted from the beads normalized by *w*[1118] negative control, determined by RNA-immunoprecipitation followed by qRT-PCR. Mean ± SD (*n* = 3). *P*-value < 0.05 (Student's *t*-test) is indicated by *. **e** ePAT assay to determine the poly-A length of *cyclin A* and *cdk1* mRNAs in control *w*[1118] and *MARF1*[null] stage 14 oocytes. The amplified DNA sizes determined by the capillary fragment analysis. Poly-A length of *cyclin A* mRNA, but not *cdk1* mRNA, is longer in *MARF1*[null] stage 14 oocytes than in control stage 14 oocytes

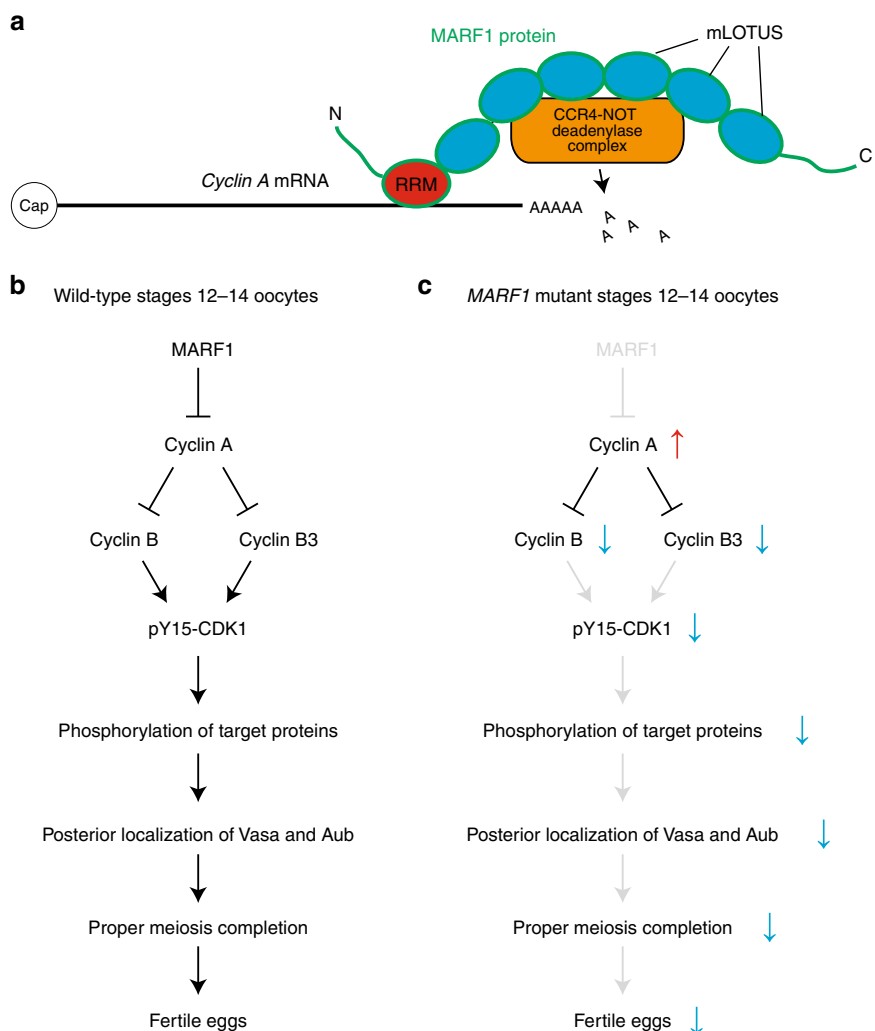

**Fig. 10** Models for MARF1 function in oocytes. **a** RRM of MARF1 binds *cyclin A* mRNA and the mLOTUS domains of MARF1 binds and recruits the CCR4-NOT deadenylase complex(es) to *cyclin A* mRNA to shorten its poly-A tail. **b** MARF1 reduces Cyclin A protein level in stages 12–14 oocytes. This leads to the protein expression of Cyclin B and Cyclin B3, which results in phosphorylation of CDK1 Tyr15. Tyr15-phosphorylated CDK1 phosphorylates many target proteins, which may promote posterior germplasm localization of Vasa and Aubergine. The proper protein phosphorylation and/or proper Vasa/Aub localization to germplasm are required for successful completion of meiosis and fertility of eggs. **c** Loss of function of *MARF1* causes an increase in Cyclin A protein level, which results in decrease of Cyclin B and Cyclin B3 protein levels in late-stage oocytes. This leads to a reduced phosphorylation of CDK1 Tyr15 and therefore an altered phosphorylation pattern of CDK1 target proteins. This may cause loss of the posterior germplasm localization of Vasa and Aubergine. Meiosis cannot complete successfully and therefore eggs are sterile

changed in *MARF1*[null] oocytes compared with control oocytes in vitro (Supplementary Fig. 4). These results indicate that in control late-stage oocytes, MARF1 post-transcriptionally regulates Cyclin A protein level by binding *cyclin A* mRNA, shortening *cyclin A* mRNA poly-A tail, and reducing Cyclin A protein level. In contrast, poly-A shortening of *cyclin A* mRNA is lost in *MARF1*[null]

oocytes, resulting in an accumulation of Cyclin A protein. Cyclin A is the only protein we found that is increased in its level in *MARF1* mutant oocytes (Fig. 5 and Supplementary Fig. 3).

Based on these findings, we propose a model for MARF1 molecular function. The MARF1 RRM binds specific target mRNAs, such as *cyclin A* mRNA, and the MARF1 mLOTUS

domains recruit the CCR4-NOT deadenylase complex (Fig. 10a). This results in shortening of target mRNA poly-A tail and reduction of protein level produced from target mRNAs. The multiple, tandem mLOTUS domain may recruit multiple CCR4-NOT deadenylase complexes per single MARF1 molecule and single target mRNA, enabling efficient poly-A shortening.

Using this model, we propose that MARF1 reduces Cyclin A protein level in stages 12–14 oocytes (Fig. 10b). This regulated reduction of Cyclin A protein level leads to expression of Cyclin B and Cyclin B3 proteins[34–38]. This cyclin proteins expressoin profile leads to stabilization of CDK1 protein and phosphorylation of CDK1 Tyr15 residue. Increase CDK1 and phosphorylation of CDK1 Tyr15 residue result in phosphorylation of appropriate target proteins of CDK1 in stage 14 oocytes. Proper global protein phosphorylation profile allows germplasm localization of Vasa and Aub and normal yolk distribution in stage 14 oocytes. As stage 14 oocytes traverse through the oviduct, meiotic Metaphase I arrest is released to complete meiosis and produce normal eggs.

Consequently, we speculate that Cyclin A is the main and/or most upstream target of MARF1. We speculate that persisted Cyclin A protein level in *MARF1* mutant late-stage oocytes arrest them in an abnormal state rather than proceeding to a normal stage 14 including decreased protein levels of Cyclin B and Cyclin B3 (Fig. 10c). Dysregulation of the three cyclin proteins levels results in the decreased CDK1 protein level and the decreased Tyr15 phosphorylation of CDK1. Dysregulation of cyclins and CDK1 alters global phosphorylation pattern. The altered phosphorylation pattern results in the loss of germplasm localization of Vasa and Aub. These together cause meiotic failure and complete sterility in *MARF1* mutants.

MARF1 seems to target specific mRNAs for gene silencing in diverse species. In this study, we identified *cyclin A* mRNA as a target of *Drosophila* MARF1. Mouse MARF1 reduces protein and mRNA levels of retrotransposons and a few endogenous genes such as PPP2CB[5,6], suggesting that they are targets of the post-transcriptional silencing by mouse MARF1. Knockdown of human MARF1 causes upregulation of IFI44L mRNA, suggesting that IFl44L mRNA a target of human MARF1[7]. Post-transcriptional gene silencing of target mRNAs in fly, mouse, and human MARF1 suggests that mLOTUS-domain directed recruitment of the CCR4-NOT deadenylase complex may be a widely conserved mechanism.

LOTUS domains are found not only in animals but also in bacteria, fungi, and plants[8,9]. LOTUS domains found in bacteria, fungi, and plants are more similar to the animal mLOTUS domains since they lack the C-terminal extension found in the animal eLOTUS domains. This suggests that the mLOTUS domain may be more ancient than the eLOTUS domain. It will be interesting to investigate the functions of these LOTUS domains found in non-animals, particularly the function of bacterial LOTUS domains, since bacteria do not have a poly-A tail in their mRNAs or CCR4-NOT deadenylase complex.

## Methods

**Fly strains**. We generated the *MARF1^null^* and *MARF1^RL1^* strains by introducing deletions within the *MARF1* coding region by genome editing using a CRISPR/Cas9 system[17,39].

The transgenic MARF1 fly strains were produced by inserting the coding sequence of MARF1 with an appropriate tag (3xHA, EGFP, lambdaN-HA, or 3xHA-HRV3Csite-3xFLAG) in a pUASPattB plasmid vector and integrating the transgenes site-specifically within the fly genome using the PhiC31 system[40,41].

**Fertility assay**. For the male fertility assay, one test male was mated with five virgin wild-type (OregonR) females in a vial[42]. After 3 days, the five OregonR females were transferred to a new vial (vial 1). Every 2 days, the five OregonR females were transferred to a new vial until a total of four vials were obtained. The five OregonR females were removed from the last vial (vial 4) after 2 days. The total number of progenies in these four vials was counted. For the female fertility assay,

five test virgin females were mated with three wild-type (OregonR) males in a cage with a 6-cm grape juice agar plate with wet yeast paste (Genesee)[42]. The grape juice agar plate was exchanged with a fresh one every day. The number of eggs laid on the third grape juice agar plate (set up on Day 3 and recovered on Day 4) was counted. Then, this grape juice agar plate was kept for one more day at 25 °C and the number of hatched eggs was counted.

**MARF1 antibodies**. We expressed a recombinant protein fragment corresponding to the 242–411 aa region of MARF1, containing the RRM and the first mLOTUS domain, as an N-terminally His-MBP-HRV3Csite-fused form using a modified pET vector in *Escherichia coli*.[43] Following purification with Ni-sepharose (GE Healthcare), the His-MBP tag was cleaved off with HRV3C protease. The MARF1 fragment was further purified using HiTrap Heparin HP (GE Healthcare). Using the purified MARF1 fragment as an antigen, we produced polyclonal anti-MARF1 sera in rabbits (Pocono Rabbit Farm & Laboratory, Inc.). The rabbit polyclonal anti-MARF1 antibodies in the sera were affinity purified using the His-MBP-HRV3Csite-MARF1 (242–411 aa) fragment described above and Affigel-15 (Bio-rad) as instructed in the product manual.

**Western blot**. Lysates of hand-dissected tissues, oocytes, unfertilized eggs, and embryos were prepared by homogenizing in RIPA buffer (50 mM Tris-HCl [pH 7.4], 150 mM NaCl, 1% [v/v] IGEPAL CA-630, 0.1% [w/v] sodium dodecyl sulfate (SDS), 0.5% [w/v] sodium deoxycholate, 1 mM ethylenediaminetetraacetic acid (EDTA), 5 mM dithiothreitol, and 0.5 mM phenylmethylsulfonyl fluoride (PMSF))[44,45]. The homogenates were clarified by centrifugation at $21,000 \times g$ at 4 °C for 10 min, and the protein concentration was determined using the BCA protein assay kit (Pierce) as needed. Fifteen micrograms of total protein was loaded per lane for Western blot. The sources and dilutions of the primary antibodies were as below. Rabbit anti-MARF1 (1/5000, generated in this study), rabbit anti-Tubulin (1/1000, Sigma, T3526), mouse anti-alpha-Tubulin (1/1500, Sigma, T9026), mouse anti-GFP (1/1000, Pierce, MA515256), mouse anti-HA (1/1000, Sigma, H3663), rat anti-Vasa (1/2000, Developmental Studies Hybridoma Bank, AB_760351), mouse anti-Cyclin A (1/1000, Developmental Studies Hybridoma Bank, AB_528188), goat anti-Cyclin A (1/1000, Santa Cruz Biotechnology, sc-15869, dN-15), mouse anti-Cyclin B (1/1000, Santa Cruz Biotechnology, sc-166152, B-6), goat anti-Cyclin B (1/1000, Santa Cruz Biotechnology, sc-15872, dN-17), rabbit anti-Cyclin B3 (1/2000,[38]), mouse anti-CDK1 (1/5000, Sigma, P7962), rabbit anti-Phospho-cdc2 (CDK1, Thr14, 1/1000, cell signaling technology, #2543), rabbit anti-Phospho-cdc2 (CDK1, Thr15, 1/1000, cell signaling technology, #9111), rabbit anti-Phospho-cdc2 (CDK1, Thr161, 1/1000, cell signaling technology, #9114), mouse anti-MPM2 (1/500, EMD Millipore, 05-368), rabbit anti-Oskar (1/3000)[46], rabbit anti-Cyclin D (1/1000, Santa Cruz Biotechnology), guinea pig anti-Png (1/2000)[16], guinea pig anti-Cort (1/2000)[47], rabbit anti-Twine (1/3000)[48], mouse anti-Aub (1/3000)[49], rabbit anti-Belle (1/2000)[50], rat anti-Nanos (1/1000)[51], rabbit anti-Me31B (1/3000)[52], mouse anti-Not1 (1/1000, 2G5)[53], rabbit anti-Not2 (1/1000, 3858)[53], rabbit anti-Not3 (1/1000)[53], rabbit anti-Caf1(1/1000, 1354)[54]. IRDye 800CW goat anti-mouse IgG, IRDye 800CW goat anti-rat IgG, IRDye 800CW goat anti-rabbit IgG, and IRDye 680RD goat anti-rabbit IgG (Licor) were used as secondary antibodies. The membranes were scanned on an Odyssey imaging system (Licor). Uncropped Western blot images are available in Supplementary Fig. 6.

**Immunostaining**. Stereomicroscope images of dissected ovaries and oocytes were taken using Leica M125 stereomicroscope.

For immunostaining of stage 14 oocytes, ovaries from 2- to 5-day-old yeast-fed and mated females were hand-dissected in Robb's buffer (55 mM NaOAc, 8 mM KOAc, 100 mM sucrose, 10 mM glucose, 1.2 mM MgCl$_2$, 1 mM CaCl$_2$, 100 mM HEPES-KOH [pH 7.4]) at room temperature, teased apart into single ovarioles. Stage 14 oocytes were separated by forceps and placed in a fixative containing 4% formaldehyde, 15 mM PIPES (pH 7.0), 80 mM KCl, 20 mM NaCl, 2 mM EDTA, and 0.5 mM ethylene glycol-bis (β-aminoethyl ether)-N,N,N′,N′-tetraacetic acid (EGTA). Fixation was done at room temperature on a rocker for 10–15 min. Fixed oocytes were rinsed three times with PBS (137 mM NaCl, 2.7 mM KCl, 1.5 mM KH$_2$PO$_4$, 8.1 mM Na$_2$HPO$_4$, pH 7.4) and dechorionated by rolling them on the frosted surface of a glass slide using a cover glass. The dechorionated oocytes were rinsed with PBS containing 0.05% (v/v) TritonX-100, extracted in PBS containing 1% (v/v) TritonX-100 at room temperature for 2 h, and rinsed with PBS containing 0.05% (v/v) TritonX-100 three times. The rinsed oocytes were incubated in blocking buffer (2% [v/v] donkey serum, 3% [w/v] BSA, 0.02% [w/v] NaN$_3$, 0.05% [v/v] TritonX-100 in PBS) for 1 h. Blocking buffer containing mouse anti-alpha-Tubulin-FITC (1/250, Sigma, F2168) was used to stain meiotic I arrested spindles in the oocytes at 4 °C overnight. DNA was stained with 0.5 µg/ml DAPI at room temperature for 15 min.

For immunostaining of unfertilized eggs, virgin female flies were placed in small fly cages with a grape juice agar plate with wet yeast paste. Eggs were collected from the plates after 2 h. The unfertilized eggs were dechorionated using 50% (v/v) bleach, rinsed with water ten times, and fixed in fixative (fixative used for stage 14 oocytes: heptone = 1:1) for 30 s with vigorous shaking and for 25 min on a rocker at room temperature. The dechorionated eggs were incubated in the same blocking buffer as used for the oocytes. Blocking buffer containing mouse anti-alpha-

Tubulin-FITC (1/250, Sigma, F2168) was used to stain polar bodies at 4 °C overnight. DNA was stained with 0.5 μg/ml DAPI at room temperature for 15 min.

For observation of the sperms that entered eggs, *w*[1118] and *MARF1*[null] virgin female flies were crossed with dj-GFP/CyO male flies[55] in fly cages. Eggs were collected after 2 h from grape juice agar plates, dechorionated, and fixed as described above.

Oocytes and eggs were mounted in Vectashield mounting reagent (Abcam). Confocal images were collected with the Zeiss LSM700 confocal microscope at the Johns Hopkins University School of Medicine Microscope Facility.

**Co-immunoprecipitation.** For protein co-immunoprecipitation, dissected ovaries of wild-type (*w*[1118]) and transgenic (MAT67Tub-Gal4 → UASP-3xHA-*MARF1*) flies were homogenized in lysis buffer (25 mM Tris-HCl [pH 7.4], 150 mM NaCl, 1 mM EDTA, 1% [v/v] IGEPAL CA-630, 5% [v/v] glycerol) containing 1× protease inhibitor cocktail; 100× protease inhibitor cocktail contains 120 mg/ml 1 mM 4-(2-aminoethyl) benzenesulfonyl fluoride hydrochloride (AEBSF), 1 mg/ml aprotinin, 7 mg/ml bestatin, 1.8 mg/ml E-64, and 2.4 mg/ml leupeptin[45]. After centrifugation, cleared supernatant was recovered. For anti-HA immunoprecipitation, the supernatant was incubated with 10% volume of Pierce anti-HA magnetic beads (Thermo Fisher Scientific) with or without 0.1 μg/ml RNase A (Thermo Fisher Scientific) at 4 °C overnight. For anti-Not1 immunoprecipitation, 2 μg of mouse monoclonal anti-Not1 antibody (2G5)[53] or purified mouse IgG (Invitrogen) were incubated with SureBeads protein G (Biorad) for 1 h at room temperature. Then the beads were washed once with lysis buffer. *w*[1118] ovary lysate supernatant was incubated with the washed beads with or without RNase A (0.1 μg/ml, Thermo Fisher Scientific) at 4 °C for 4 h. After washing five times with lysis buffer, proteins were eluted with 2× SDS-PAGE loading buffer (80 mM Tris-HCl [pH 6.8], 2% [w/v] SDS, 10% [v/v] glycerol, 0.0006% [w/v] bromophenol blue, 2% [v/v] 2-mercaptoethanol) at 95 °C for 5 min.

For RNA co-immunoprecipitation, hand-dissected ovaries of *w*[1118] and MAT67Tub-Gal4 → 3xHA-HRV3Csite-3xFLAG-MARF1 flies were homogenized in lysis buffer containing RNase inhibitor (50 U/μl, Thermo Fisher Scientific). After centrifugation, cleared supernatant was incubated with 10% volume of Pierce anti-HA Agarose (Thermo Fisher Scientific) at 4 °C for 2 h. After washing five times with the lysis buffer, 3xFLAG-MARF1 protein–RNA complex was released by HRV3C protease cleavage in cleavage solution (25 mM Tris-HCl [pH 7.4], 150 mM NaCl, 5% [v/v] glycerol, 2 mM EDTA, 200 nM HRV3C protease) at 4 °C overnight. RNA was purified from the supernatant using Trizol LS (Sigma).

**ePAT.** RNA from oocytes was prepared using miRVana (Thermo Fisher Scientific). ePAT assay[26] was performed using the primers listed in Supplementary Table 1. For the capillary fragment analysis, forward primers with 6-fluorescein amidite (FAM) were used. The fragment analysis data were analyzed using Peak Scanner 2 (Thermo Fisher Scientific).

**qRT-PCR.** RNA from oocytes was prepared using miRVana (Thermo Fisher Scientific). RNAs extracted from oocytes and RNAs coimmunoprecipitated with proteins were treated with Turbo DNase (Thermo Fisher Scientific), and then were reverse-transcribed into cDNA using a random hexamer primer and AMV Reverse Transcriptase (NEB). qPCR was performed using iTaq Universal SYBR Green Supermix or SsoAdvanced Universal SYBR Green Supermix on CFX96 (Biorad). The primers used are listed in Supplementary Table 1.

**In vitro Cyclin A protein degradation assay.** 6xHis-MBP-HRV3Csite-HA-Cyclin A protein was expressed using a modified pET plasmid vector in *E. coli*[43]. The protein was purified using Ni-sepharose beads (GE Healthcare). The protein was incubated with 6xHis-HRV3C protease at 4 °C overnight to cleave off the 6xHis-MBP fragment from the CycA protein. The cleaved product, HA-CycA, was further purified by passing through Ni-sepharose, buffer-exchange, and concentration using Amicon-Ultra-15 centrifugal unit (EMD Millipore). The purified HA-CycA protein in buffer (20 mM Hepes-NaOH [pH 8.0], 500 mM NaCl, 0.25 mM TCEP, and 50% glycerol) was flash-frozen in liquid nitrogen and stored at −80 °C until use.

Hand-dissected stage 14 oocytes were homogenized in lysis buffer (30 mM Hepes-KOH [pH 7.4], 100 mM KOAc, 2 mM MgOAc), followed by protein concentration determination with BCA protein assay kit (Pierce). Then final 1 mM DTT was added to the lysates. About 200 ng of HA-Cyclin was incubated in the stage 14 oocyte lysates (about 540 ng total protein) in the presence of 1 mM ATP in 100 μl scale at 25 °C. Ten microliters of the reaction mixture was taken at each time point, mixed with 2× SDS PAGE loading buffer, and heated at 95 °C for 5 min for Western blot analysis. Half-life ($t_{1/2}$) was determined by curve fitting of an equation —Fraction remaining = $(1/2)^{\wedge}(\text{time}/t_{1/2})$—using Igor Pro 6.31 (Wavematrix)[56].

**mRNA sequencing.** poly-A+ mRNA libraries were prepared and sequenced on Hiseq4000 as in manufacture protocols (Illumina)[42,57,58].

## Data availability

SRA accession number for the mRNA libraries reported in this paper is SRP133238.

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

## Acknowledgements

We thank Dr. Julius Brennecke (Institute of Molecular Biotechnology, Vienna) for his gifts of fly strains. We thank Dr. Anne Ephrussi (European Molecular Biology Laboratory), Dr. Terry Orr-Weaver (Massachusetts Institute of Technology), Dr. Patrick O'Farrell (University of California, San Francisco), Dr. Paul Macdonald (University of Texas at Austin), Dr. Elmar Wahle (Martin Luther University Halle-Wittenberg), Dr. Christian Lehner (University of Zurich), Dr. Siomi Haruhiko (Keio University), Dr. Akira Nakamura (Kumamoto University), and Dr. Toshie Kai (Osaka University) for their kind gifts of antibodies. We thank Bloomington *Drosophila* Stock Center, *Drosophila* Genomics and Genetic Resources, Kyoto Stock Center, and Vienna *Drosophila* Resource Center for fly strain stocks. We thank the Johns Hopkins University School of Medicine Microscope Facility for use of the Zeiss LSM700, supported by NIH grant S10OD016374 awarded to Dr. Scot C. Kuo. This work was supported by the grants from American Heart Association [15SDG23220028] and the National Institutes of Health [R01GM116841] to R.F.

## Author contributions

Conceptualization, L.Z. and R.F.; Methodology, L.Z., S.K.K. and R.F.; Investigation, L.Z., S.K.K., S.E.L., and R.F. Writing–original draft, L.Z. and R.F.; Writing–review and editing, L.Z., S.K.K., S.E.L. and R.F.; Funding acquisition, R.F.; Supervision, R.F.

## Additional information

**Competing interests:** The authors declare no competing interests.

