## [Peer Review File · Nature Communications]

Reviewers' comments:

Reviewer #1 (Remarks to the Author):

This manuscript from Li, et al. examines the role of MARF1, a LOTUS-domain-containing protein, in *Drosophila* oogenesis. Although LOTUS domains of other proteins (such as Tejas and Oskar) interact with Vasa, the roles of the MARF1 LOTUS domain and of MARF1 are unknown. MARF1 mutant females do not produce viable embryos, likely because of meiotic defects. The authors demonstrate via tethering assays that MARF1 is a post-transcriptional repressor that most likely acts by recruiting the CCR4-NOT deadenylase complex to destabilize or translationally repress targets. The authors suggest that cyclin A is a direct target of MARF1, and its derepression in MARF1 mutants is responsible for many of the observed phenotypes. Although their phenotypic characterization of MARF1 is compelling, the authors' molecular analyses are less so. My concerns diminish my enthusiasm for the manuscript in its current form.

Major comments:

- 1) The authors state that "cyclin A is the main and/or most upstream target of MARF1." However, changes in cyclin B/B3 and CDK1 precede changes in cyclin A expression, an observation that is contrary to cyclin A being the main target. In addition, Vasa and Aub localization change in MARF1 mutants, suggesting pleiotropic effects. Is there evidence that the changes in cyclin B levels depend on derepression of cyclin A in the MARF mutants? Does MARF1 bind cyclin B/B3 mRNA?
- 2) Formally, the prolonged expression of cyclin A in stage 14 oocytes could be explained by increased translation (a model the authors favor) or by decreased degradation of the cyclin A protein. Is there direct evidence that cyclin A continues to be translated in stage 14 MARF^{-/-} oocytes?
- 3) A critical question is whether other mRNAs are also target of MARF. Are there changes in RNA abundance in MARF mutant oocytes? Or changes in translation (as would be predicted based on the cyclin A results)? Is there any effect on transposon expression?
- 4) On pg 15, the authors state that "MARF1, when tethered to the 3'UTR of a reporter RNA, suppresses protein expression from the mRNA." Given that mRNA abundance also decreases, a more likely conclusion is that the majority of the effect of MARF1, when tethered, is explained by mRNA decay, rather than translational inhibition.
- 5) In the tethering experiments, the authors observe a decrease in mRNA abundance when MARF1 was tethering; however, for the proposed endogenous target (cyclin A), there was no difference in abundance. What happens if a reporter containing the cyclin A UTRs is used? Could these differences be explained by precocious expression of the NHA-MARF1 construct, compared to the endogenous version?

Minor comments

- 1) mRNA high-throughput sequencing is listed in the materials and methods, but there is no

mention of these libraries in the main text.

2) Is the loss of MARF1 at the oocyte-to-embryo transition PNG dependent?

Reviewer #2 (Remarks to the Author):

Zhu and colleagues investigate the function of MARF-1 a conserved protein with RNA binding (RRM) and LOTUS domains. They study its role in *Drosophila*. Some of the new results in this paper are interesting but others are not convincing.

First, the authors generate an antibody and show that MARF1 is only found in late oocytes. They say that it is not found in stage 1-11 oocytes or in unfertilized eggs, embryos, or males, although weak signal is visible in their Stage 11 oocyte lane. They report that an EGFP fusion to MARF is cytoplasmic and nuclear, but they present no evidence that this fusion protein is functional. Thus the results on expression and localization are only partially convincing.

Next, the authors make an apparent null allele and a truncation allele of MARF1. On Western blots they do not detect the small peptide encoded by their null allele, but this not surprising since the antibody they use is not directed against this part of the protein. So although it is likely that the allele is null, their molecular evidence for this is not meaningful. They detect the truncated protein, which is smaller than the endogenous protein, but they do not say whether the protein is of the expected size. They include in Fig. 1 a HA-tagged MARF1 which they say is produced "at physiological levels", contradicting what they later say about this tagged protein in Fig. 7. The Western results thus are partially but not fully convincing.

Next the authors show that MARF1 mutant females are sterile. This was convincing, although the sample size was very small ($n=3$). With an unstated sample size, they conclude that MARF1 null eggs can be fertilized.

Next, the authors report abnormalities in MARF1 mutant oocytes, including in yolk, in polar body formation, in levels or positions of certain proteins. It was surprising that yolk was abnormal in a mutant for an RNA binding protein. The simplest explanation is that MARF1 arrests oogenesis, and that yolk abnormality reflects that. This same model could imply that other molecular effects of the mutant including lack of meiotic spindle, lack of posterior localized Aub and consequences on Cyclin levels are also indirect, reflecting oogenesis arrest, possibly followed by abnormal proceeding by the arrested oocyte. The authors report that MARF1 unfertilized eggs have no polar body. They show a small part of the egg from one of their few samples, showing no obvious polar body. But then, where did the oocyte nucleus go?

The authors report that MARF1 can function in some assays and can rescue the sterility, but two truncated versions cannot. However, in the tethering assay, an interesting part of the paper, a truncation that retains one LOTUS domain can function. The authors propose that this shows function for the LOTUS domain, but they do not rule out that truncated version

that did not function in the assay was not simply unstable or mis-folded. This makes the conclusion unconvincing. Also in the tethering experiment the authors note that MARF1 binding correlates with shortened poly-A tail and decreased stability of the target RNA whereas GW182 control's binding also shortens the poly-A tail but does not affect the RNA's stability. It was hard to understand why the two results differed, as shortened poly-A tails normally destabilize mRNAs unless MARF1 has an additional untested function.

In the IPs with MARF1-HA, members of the CCR4-NOT deadenylase complex precipitate. This is the other interesting result in the paper, and may be the most convincing. But their statements in the text do not always match what the data show. For example, endogenous MARF1 was precipitated in the +RNase late of Fig. 7A even though the text on p. 17 says it isn't.

Other:

The paper needs editing. Some phrases are ungrammatical. Also, in the introduction and discussion in particular, the authors often state the same thing multiple times in different ways, which makes those sections much longer than needed and makes it hard to keep the point of the section in mind. Finally, a number of statements in those sections lack necessary references, or citations to primary literature.

The authors say that Cyclin A is the only protein whose level increases in MARF1 oocytes. But it appears from Fig. 4 that they only looked at five proteins, plus some unclear MPM2 reactive bands. Many other proteins may be affected. This weakens their model that "Cyclin A is the main target of MARF1".

The authors give too little information to evaluate or reproduce the work. For example, how much protein is loaded in the Western blots? Were the experiments repeated? Sample sizes are missing or very low, as noted earlier.

We performed the following additional experiments to address reviewers' comments. Experiments are listed in the order in which they appear in the manuscript.

1. Western blot to examine MARF1 protein level in *png* mutant stage 14 oocytes and early embryos (Fig. 1C) to address Reviewer 1's question on whether the rapid decay of MARF1 at the oocyte-to-embryo transition is PNG dependent.

2. Western blot to examine CycA protein level in *MARF1* mutant oocytes and eggs with an increased sample size (Fig. 4) to address Reviewer 1's question on its change in *MARF1* mutant stage 12-13 oocytes.

3. qRT-PCR to measure GFP mRNA level in MARF1 tethering assay with an increased sample size (Fig. 5G) to address Reviewer 1 and 2's question whether decreased protein level is due to poly-A tail shortening or mRNA decay.

4. Co-IP followed by Western blot to examine interaction between HA-MARF1 and endogenous MARF1, GW-182, and Ge-1 (Fig. 7A) to address Reviewer 2's question whether HA-MARF1 binds endogenous MARF1 and to show the specific interaction between MARF1 and the CCR4-Not complex more convincingly.

5. Co-IP followed by Western blot to examine interaction between endogenous Not1 and endogenous MARF1 (Fig. 7B) to further strengthen the most interesting point of the paper; MARF1 binds the CCR4-Not complex.

6. RNA-IP followed by qRT-PCR to test binding of HA-MARF1 to *cyclin A/B/B3* mRNAs (Fig. 7D) to address Reviewer 1's question whether MARF1 binds *cyclin B/B3* mRNAs.

7. Fertility assay to confirm that EGFP-MARF1 is fully functional and can rescue female fertility of *MARF1^{null}* (Fig. S1A-C) to address Reviewer 2's question whether EGFP-MARF1 is functional.

8. Confocal microscopy imaging of EGFP-MARF1 with an appropriate negative control to examine background level (Fig. S1D) to address Reviewer 2's question on localization pattern of EGFP-MARF1.

9. Western blot to examine Oskar protein level in *MARF1* mutant oocytes (Fig. S4) to address Reviewer 2's question on proteins other than cyclins that are dysregulated in *MARF1* mutant oocytes.

10. In vitro Cyclin A degradation assay in *MARF1^{null}* and control stage 14 oocyte lysates (Fig. S5) to address Reviewer 1's question whether prolonged expression of Cyclin A in *MARF1* mutant oocytes is due to an increased production or decreased degradation of Cyclin A protein.

11. GFP reporter assay to test effects of transgenic HA-MARF1 on GFP protein level produced from GFP mRNA containing *cyclin A/B/B3* UTRs (Fig. R1) to address Reviewer 1's question whether a reporter mRNA containing *cyclin A* UTRs is regulated by transgenic HA-MARF1.

Our detailed answers to the reviewers' comments are listed below.

We also revised the text to address the reviewers' concerns where appropriate.

We hope that this revised manuscript is now acceptable for publication in *Nature Communications*.

Reviewer #1 (Remarks to the Author):

This manuscript from Li, et al. examines the role of MARF1, a LOTUS-domain-containing protein, in *Drosophila* oogenesis. Although LOTUS domains of other proteins (such as Tejas and Oskar) interact with Vasa, the roles of the MARF1 LOTUS domain and of MARF1 are unknown. MARF1 mutant females do not produce viable embryos, likely because of meiotic defects. The authors demonstrate via tethering assays that MARF1 is a post-transcriptional repressor that most likely acts by recruiting the CCR4-NOT deadenylase complex to destabilize or translationally repress targets. The authors suggest that cyclin A is a direct target of MARF1, and its derepression in MARF1 mutants is responsible for many of the observed phenotypes. Although their phenotypic characterization of MARF1 is compelling, the authors' molecular analyses are less so. My concerns diminish my enthusiasm for the manuscript in its current form.

Major comments:

1) The authors state that “cyclin A is the main and/or most upstream target of MARF1.” However, changes in cyclin B/B3 and CDK1 precede changes in cyclin A expression, an observation that is contrary to cyclin A being the main target. In addition, Vasa and Aub localization change in MARF1 mutants, suggesting pleiotropic effects. Is there evidence that the changes in cyclin B levels depend on derepression of cyclin A in the MARF mutants? Does MARF1 bind cyclin B/B3 mRNA?

In the previous draft, a large variation was observed among biologically replicated experiments of Western blot testing CycA level in stages 12-13 oocytes. Thus, we did not observe a statistically significant difference in CycA level in *MARF1* mutant. In the revised manuscript, we repeated the Western blot experiments with an increased number of biological replicates. Now we detected a statistically significant increase ($p < 0.05$) in the CycA protein level in *MARF1* mutant stages 12-13 oocytes compared with control stages 12-13 oocytes (Fig. 4). Therefore, changes in Cyclin B/B3 and CDK1 do not precede changes in Cyclin A.

Furthermore, we showed that MARF1 does not bind *cyclin B/B3* mRNAs (Fig. 7D).

These results support our model that *cyclin A* mRNA is the main target of MARF1.

2) Formally, the prolonged expression of cyclin A in stage 14 oocytes could be explained by increased translation (a model the authors favor) or by decreased degradation of the cyclin A protein. Is there direct evidence that cyclin A continues to be translated in stage 14 *MARF1*^{-/-} oocytes?

We tested if degradation of Cyclin A protein is decreased in *MARF1* mutant stage 14 oocytes. We prepared recombinant HA-Cyclin A protein and *MARF1* mutant and control stage 14 oocyte lysates. We performed recombinant HA-Cyclin A protein degradation assay in vitro using the lysates. We found that HA-Cyclin A protein is degraded at similar rates in *MARF1* mutant and control stage 14 oocyte lysates. This result supports our

model that increased translation, rather than decreased degradation, of Cyclin A protein causes the prolonged expression of Cyclin A in *MARF1* mutant stage 14 oocytes.

We also considered directly testing whether Cyclin A translation is increased using polysome profiling and/or ribosome profiling. However, we found that it is technically challenging due to the large number hand-dissected stage 14 oocytes required for this experiment.

3) A critical question is whether other mRNAs are also target of MARF. Are there changes in RNA abundance in MARF mutant oocytes? Or changes in translation (as would be predicted based on the cyclin A results)? Is there any effect on transposon expression?

We measured mRNA abundance in *MARF1* mutant and control oocytes. We found no change in mRNA levels of *cyclin A/B/B3*, *cdk1*, *vasa*, *aubergine*, *belle*, *oskar*, etc (Fig. S5). There is no change in transposon mRNA either. We found several mRNAs that are differentially expressed in *MARF1* mutant oocytes, but none of them clearly explains the sterile phenotype of *MARF1* mutants. We tested ~20 proteins by Western blot. Among these ~20 proteins, CycA, CycB, and CycB3, and CDK1 were the only proteins that showed differential expression (Fig.4, Fig S4, and not shown).

4) On pg 15, the authors state that “MARF1, when tethered to the 3’UTR of a reporter RNA, suppresses protein expression from the mRNA.” Given that mRNA abundance also decreases, a more likely conclusion is that the majority of the effect of MARF1, when tethered, is explained by mRNA decay, rather than translational inhibition.

We carefully repeated the qRT-PCR experiments to measure the GFP reporter mRNA level with an increased number of biological replicates. We did not observe any statistically significant change in mRNA level caused by MARF1 tethering (Fig. 5G). This result supports our conclusion that majority of the MARF1 effect is explained by translational inhibition (caused by poly-A tail shortening) rather than mRNA decay.

5) In the tethering experiments, the authors observe a decrease in mRNA abundance when MARF1 was tethering; however, for the proposed endogenous target (cyclin A), there was no difference in abundance. What happens if a reporter containing the cyclin A UTRs is used? Could these differences be explained by precocious expression of the NHA-MARF1 construct, compared to the endogenous version?

[redacted]

Minor comments

1) mRNA high-throughput sequencing is listed in the materials and methods, but there is no mention of these libraries in the main text.

mRNA-seq results are discussed in the text (pages 13-14) and are shown in Figure S5.

2) Is the loss of MARF1 at the oocyte-to-embryo transition PNG dependent?

We examined MARF1 protein level in two *png* mutants. While MARF1 protein level is not changed in *png* mutant stage 14 oocytes, MARF1 protein level is increased in *png* mutant 0-2 hr embryos (Fig. 1C). Therefore, we conclude that the loss of MARF1 at the oocyte-to-embryo transition is PNG-dependent. We thank the reviewer very much for the insightful question, which helped us to make this very interesting finding. The finding is described in page 5.

Reviewer #2 (Remarks to the Author):

Zhu and colleagues investigate the function of MARF-1 a conserved protein with RNA binding (RRM) and LOTUS domains. They study its role in *Drosophila*. Some of the new results in this paper are interesting but others are not convincing.

First, the authors generate an antibody and show that MARF1 is only found in late oocytes. They say that it is not found in stage 1-11 oocytes or in unfertilized eggs, embryos, or males, although weak signal is visible in their Stage 11 oocyte lane. They report that an EGFP fusion to MARF is cytoplasmic and nuclear, but they present no evidence that this fusion protein is functional. Thus the results on expression and localization are only partially convincing.

We edited the text to make it clear that there is a weak expression of MARF1 protein in stage 1-11 oocytes. (page 5)

We examined if the transgenic EGFP-MARF1 is functional by testing if it can rescue female fertility and stage 14 oocytes morphology of *MARF1^{null}* mutants. Germline-expression of the transgenic EGFP-MARF1 fully rescued the female fertility and stage 14 oocytes morphology of *MARF1^{null}* mutants (Fig. S1). Furthermore, we also tested flies with germline-expression of 3xHA-MARF1 or EGFP-MARF1 in the presence of endogenous MARF1. We found that these flies show similar fertility to the flies without such transgenic MARF1 proteins. Thus EGFP-MARF1, as well as 3xHA-MARF1, is functional.

For the EGFP-MARF1 expression and localization, we confirmed that EGFP signals in negative controls that do not express EGFP-MARF1 are much lower than those in oocytes expressing EGFP-MARF1 (Fig. S1D). Therefore, we conclude that the strong EGFP signal we observed in oocytes expressing EGFP-MARF1 are derived from EGFP-MARF1, not background.

Next, the authors make an apparent null allele and a truncation allele of MARF1. On Western

blots they do not detect the small peptide encoded by their null allele, but this not surprising since the antibody they use is not directed against this part of the protein. So although it is likely that the allele is null, their molecular evidence for this is not meaningful. They detect the truncated protein, which is smaller than the endogenous protein, but they do not say whether the protein is of the expected size. They include in Fig. 1 a HA-tagged MARF1 which they say is produced “at physiological levels”, contradicting what they later say about this tagged protein in Fig. 7. The Western results thus are partially but not fully convincing.

The truncated MARF1 protein observed from the MARF1^{RL1} allele has the expected size (46 kDa). This is now mentioned in the text. (page 7)

The very short 34 aa peptide derived from the N-terminus of MARF1 protein, if produced from the MARF1^{null} allele and exists as a stable form, is only 4 kDa in size. It's too small to be detected by SDS-PAGE-Western blot, even if we try by raising an antibody that detects this fragment. Such short peptide, compared with the full-length 1305 aa protein, is not likely to have any functions.

We used different Gal4 drivers between Fig. 1 (MAT15-Gal4 on the third chromosome) and Fig. 7 (MAT67-Gal4 on the second chromosome). It is well known that different Gal4 drivers express the same UAS transgene differently. So there is nothing self-contradictory in our data. We also would like to point out that HA-MARF1 protein level is slightly lower than the endogenous MARF1 in Fig. 7A. This is a more favorable outcome than overexpression of a transgenic protein, which avoids potential non-physiological protein-protein interaction caused by overexpression.

Next the authors show that MARF1 mutant females are sterile. This was convincing, although the sample size was very small (n=3). With an unstated sample size, they conclude that MARF1 null eggs can be fertilized.

As described in the Materials and Methods section, in the female fertility assay, 5 females were tested in each cage. We showed the number of cages as n. So, n=3 means we tested 15 females. None of the 15 females of MARF1 mutant produced hatchable eggs in Figure 2C, for example. Moreover, the MARF1 mutant flies were tested repeatedly and independently in different assays and always showed 0% hatching rate (Fig. 2C, S1B, S2D, and S2G). Therefore, the results are extremely highly reproducible. In the revised manuscript, we clearly stated the sample size in each experiment (including Fig. S2A, for which the reviewer asked the sample size). We also plotted individual data points in all graphs, instead of showing just means and errors.

Next, the authors report abnormalities in MARF1 mutant oocytes, including in yolk, in polar body formation, in levels or positions of certain proteins. It was surprising that yolk was abnormal in a mutant for an RNA binding protein. The simplest explanation is that MARF1 arrests oogenesis, and that yolk abnormality reflects that. This same model could imply that other molecular effects of the mutant including lack of meiotic spindle, lack of posterior localized Aub and consequences on Cyclin levels are also indirect, reflecting oogenesis arrest, possibly followed by abnormal proceeding by the arrested oocyte. The authors report that MARF1 unfertilized eggs

have no polar body. They show a small part of the egg from one of their few samples, showing no obvious polar body. But then, where did the oocyte nucleus go?

Nucleus was not observed in *MARF1* mutant eggs, suggesting that they were disintegrated.

The authors report that *MARF1* can function in some assays and can rescue the sterility, but two truncated versions cannot. However, in the tethering assay, an interesting part of the paper, a truncation that retains one LOTUS domain can function. The authors propose that this shows function for the LOTUS domain, but they do not rule out that truncated version that did not function in the assay was not simply unstable or mis-folded. This makes the conclusion unconvincing. Also in the tethering experiment the authors note that *MARF1* binding correlates with shortened poly-A tail and decreased stability of the target RNA whereas GW182 control's binding also shortens the poly-A tail but does not affect the RNA's stability. It was hard to understand why the two results differed, as shortened poly-A tails normally destabilize mRNAs unless *MARF1* has an additional untested function.

We examined the level of the truncated transgenic *MARF1* proteins in the ovary lysates, after removing insoluble fractions by high speed centrifugation. Strong band signals were observed for the truncated proteins (actually stronger than the full-length transgenic *MARF1*) (Fig. 6C and 7C). Furthermore, these truncated proteins were efficiently immunoprecipitated by anti-HA beads, showing that the proteins are soluble and their HA-tag is accessible. These results indicate that the truncated proteins are stable and folded.

As in the response to the reviewer 1 point (4), we observed no change in the reporter GFP mRNA level in *MARF1* tethering as well as GW182 tethering (Fig. 5G). Thus, the two results are similar.

In the IPs with *MARF1*-HA, members of the CCR4-NOT deadenylase complex precipitate. This is the other interesting result in the paper, and may be the most convincing. But their statements in the text do not always match what the data show. For example, endogenous *MARF1* was precipitated in the +RNase late of Fig. 7A even though the text on p. 17 says it isn't.

Since the Western blot image in the previous manuscript version was not clear enough to make a conclusion, we repeated the experiments. Now it is clearer that there is no endogenous *MARF1* band that was co-precipitated with HA-*MARF1* (Fig. 7A).

To make the most interesting and novel finding of the paper ("MARF1 binds the CCR4-NOT deadenylase complex"), more convincing, we additionally tested whether endogenous Not1 and endogenous *MARF1* bind. We immunoprecipitated endogenous Not1 and found that endogenous *MARF1* was co-precipitated with or without RNase A (Fig. 7B). Thus we conclude that endogenous Not1 and *MARF1* bind independent of RNA.

Other:

The paper needs editing. Some phrases are ungrammatical. Also, in the introduction and

discussion in particular, the authors often state the same thing multiple times in different ways, which makes those sections much longer than needed and makes it hard to keep the point of the section in mind. Finally, a number of statements in those sections lack necessary references, or citations to primary literature.

We removed redundant statements, corrected grammatical errors, and added additional necessary references or citations to primary literature.

The authors say that Cyclin A is the only protein whose level increases in MARF1 oocytes. But it appears from Fig. 4 that they only looked at five proteins, plus some unclear MPM2 reactive bands. Many other proteins may be affected. This weakens their model that “Cyclin A is the main target of MARF1”.

By testing levels of some additional proteins such as Oskar by the Western blot, we now examined ~ 20 proteins in total in Fig. 4 and S4 (and not shown). Still, Cyclin A is the only increased protein, supporting our model that Cyclin A is the main target of MARF1. Importantly, we also showed specific binding of MARF1 to *cyclin A* mRNA (Fig. 7), further supporting our model.

The authors give too little information to evaluate or reproduce the work. For example, how much protein is loaded in the Western blots? Were the experiments repeated? Sample sizes are missing or very low, as noted earlier.

In the updated Materials and Methods section and Figure legends, we clearly stated the loaded protein amounts, sample sizes, and whether the experiments were repeated, We also increased the sample size for some of the experiments compared to the previous version of the manuscript.

REVIEWERS' COMMENTS:

Reviewer #1 (Remarks to the Author):

This revised manuscript from Zhu et al. is greatly strengthened with the additional experiments, and my concerns have been addressed. I support publication of this manuscript and know that it will be of interest to many in the field.

One minor comment: on pg 14, the authors write that "GW182 is a known translational repressor." GW182 is known both to repress translation and to stimulate mRNA decay (at least in S2 cells), and the distinction between its effect in S2 cells and in oocytes should be made in the text.

Reviewer #2 (Remarks to the Author):

The authors did an excellent job adding data, information, and clarifications to the paper. It is now clear, convincing, and novel. I am impressed with their thorough revisions. I recommend publication after the following straightforward modifications are made:

a. I remain uneasy about the disappearance of the nucleus, as well as the abnormalities in yolk proteins seen in the MARF1 mutant oocytes. The yolk abnormality, for example, cannot be explained simply by abnormal cyclin A levels. The authors' only response was that they think the nuclei disintegrated. I agree that that probably happened, but my concern had been about the reason for this. I believe that another possible explanation is that the mutation, rather than only affecting cyclin levels, caused oogenesis to arrest in an abnormal state, rather than proceeding to a normal stage 14. I urge the authors to include a statement about this possibility. It does not negatively impact any of their conclusions.

b. On p. 5 the authors state that there is no MARF1 in 0-2hr fertilized eggs, as in Fig. 1B, but later that page, and in Fig. 1C, they say that there is a little MARF1 detected in the wildtype (png/+) controls. A small rephrasing at the first mention would fix this issue.

c. The wording of the second sentence of the section that begins on p. 8 needs slight modification. Failure of laid MARF1- eggs to develop could be due to a problem in oocyte maturation as stated or to a problem immediately afterwards, during egg activation. These can be distinguished, and the authors did so and can say that: their finding of yolk protein and cyclin A abnormalities in stage 14 oocytes shows clearly that it is a maturation defect.

d. Addition of poly A tails also factors into proteins' translation. Work on the wispy poly A polymerase (Benoit et al. 2008, Cui et al. 2008, 2013) in *Drosophila* oocytes is relevant to consider in the paper and model.

e. p. 21 top: please add the word "target" in front of "mRNAs". The paper clearly shows that the effect is not general to all mRNAs, as would be implied by the sentence without

“target”.

f. At a few places in the text, wording or grammar needs attention to be in idiomatic English.

Not for this paper, just a suggestion for the authors: for future experiments (not needed for this paper) it will be interesting to see if 1-340 can rescue the fertility phenotype (e.g. is one mLOTUS domain enough or are multiple better, as hypothesized in the Discussion?).

REVIEWERS' COMMENTS:

Reviewer #1 (Remarks to the Author):

This revised manuscript from Zhu et al. is greatly strengthened with the additional experiments, and my concerns have been addressed. I support publication of this manuscript and know that it will be of interest to many in the field.

One minor comment: on pg 14, the authors write that "GW182 is a known translational repressor." GW182 is known both to repress translation and to stimulate mRNA decay (at least in S2 cells), and the distinction between its effect in S2 cells and in oocytes should be made in the text.

We edited the text. "GW182 represses translation and destabilizes mRNAs in S2 cells while it represses translation without changing mRNA levels in oocytes³²." (page 14).

Reviewer #2 (Remarks to the Author):

[Previous reviewer #2.]

The authors did an excellent job adding data, information, and clarifications to the paper. It is now clear, convincing, and novel. I am impressed with their thorough revisions. I recommend publication after the following straightforward modifications are made:

a. I remain uneasy about the disappearance of the nucleus, as well as the abnormalities in yolk proteins seen in the MARF1 mutant oocytes. The yolk abnormality, for example, cannot be explained simply by abnormal cyclin A levels. The authors' only response was that they think the nuclei disintegrated. I agree that that probably happened, but my concern had been about the reason for this. I believe that another possible explanation is that the mutation, rather than only affecting cyclin levels, caused oogenesis to arrest in an abnormal state, rather than proceeding to a normal stage 14. I urge the authors to include a statement about this possibility. It does not negatively impact any of their conclusions.

We added the statement "Consequently, we speculate that Cyclin A is the main and/or most upstream target of MARF1 and that persisted Cyclin A protein level in *MARF1* mutant late stage oocytes arrest them in an abnormal state rather than proceeding to a normal stage 14 including decreased protein levels of Cyclin B and Cyclin B3 (Fig. 10C)." (page 21).

b. On p. 5 the authors state that there is no MARF1 in 0-2hr fertilized eggs, as in Fig. 1B, but later that page, and in Fig. 1C, they say that there is a little MARF1 detected in the wildtype (png/+) controls. A small rephrasing at the first mention would fix this issue.

We fixed it. "MARF1 is only very weakly expressed in stages 1-11 oocytes and 0-2 hr embryos and is not detected in unfertilized eggs, whole male flies, the carcasses of female flies after ovary dissection, or S2 cells." (page 5)

c. The wording of the second sentence of the section that begins on p. 8 needs slight modification. Failure of laid MARF1- eggs to develop could be due to a problem in oocyte maturation as stated or to a problem

immediately afterwards, during egg activation. These can be distinguished, and the authors did so and can say that: their finding of yolk protein and cyclin A abnormalities in stage 14 oocytes shows clearly that it is a maturation defect.

We edited. "This may be due to defective oocyte maturation or defects during egg activation." (page 8) and then "This shows that MARF1 is required for proper oocyte maturation." (page 8).

d. Addition of poly A tails also factors into proteins' translation. Work on the wispy poly A polymerase (Benoit et al. 2008, Cui et al. 2008, 2013) in *Drosophila* oocytes is relevant to consider in the paper and model.

We added a sentence "Regulating mRNA poly-A tail length is important for oocyte maturation. For example, addition of poly-A tails to mRNAs by the Wispy cytoplasmic poly-A polymerase is one of the important post-transcriptional mechanisms¹⁹⁻²¹" and cited the papers the reviewer suggested. (page 18)

e. p. 21 top: please add the word "target" in front of "mRNAs". The paper clearly shows that the effect is not general to all mRNAs, as would be implied by the sentence without "target".

We added "target". (page 20)

f. At a few places in the text, wording or grammar needs attention to be in idiomatic English.

We corrected wording and grammars.

Not for this paper, just a suggestion for the authors: for future experiments (not needed for this paper) it will be interesting to see if 1-340 can rescue the fertility phenotype (e.g. is one mLOTUS domain enough or are multiple better, as hypothesized in the Discussion?).

We already had this data (Supplementary Figure 2F-G). The 1-340 aa fragment cannot rescue the fertility phenotype.